# Manipulation of the human tRNA pool reveals distinct tRNA sets that act in cellular proliferation or cell cycle arrest

Noa Aharon-Hefetz[1], Idan Frumkin[1], Yoav Mayshar[2], Orna Dahan[1]*, Yitzhak Pilpel[1]*, Roni Rak[1]

[1]Department of Molecular Genetics, Weizmann Institute of Science, Rehovot, Israel; [2]Department of Molecular Cell Biology, Weizmann Institute of Science, Rehovot, Israel

**Abstract** Different subsets of the tRNA pool in human cells are expressed in different cellular conditions. The 'proliferation-tRNAs' are induced upon normal and cancerous cell division, while the 'differentiation-tRNAs' are active in non-dividing, differentiated cells. Here we examine the essentiality of the various tRNAs upon cellular growth and arrest. We established a CRISPR-based editing procedure with sgRNAs that each target a tRNA family. We measured tRNA essentiality for cellular growth and found that most proliferation-tRNAs are essential compared to differentiation-tRNAs in rapidly growing cell lines. Yet in more slowly dividing lines, the differentiation-tRNAs were more essential. In addition, we measured the essentiality of each tRNA family upon response to cell cycle arresting signals. Here we detected a more complex behavior with both proliferation-tRNAs and differentiation tRNAs showing various levels of essentiality. These results provide the so-far most comprehensive functional characterization of human tRNAs with intricate roles in various cellular states.

**\*For correspondence:**
Orna.Dahan@weizmann.ac.il (OD); Pilpel@weizmann.ac.il (YP)

**Competing interests:** The authors declare that no competing interests exist.

## Introduction

Cells in multicellular species may typically exist in one of two alternative states, they either proliferate or they are cell cycle-arrested (*Cooper, 2000*). Differentiated cells are typically less proliferative or they may not divide at all, while proliferation occurs often prior to terminal differentiation, or when differentiation is reversed, predominantly in cancer (*Hanahan and Weinberg, 2011*; *Ruijtenberg and van den Heuvel, 2016*; *Hafner et al., 2019*). The proliferation rate ranges from fast dividing cells (e.g. hematopoietic stem cells), through quiescent cells that only divide upon a need to replace dead or injured cells (e.g. fibroblasts, smooth muscle cells, epithelial cells), to cells with little or no proliferation potential (e.g. cardiac muscle tissue) (*Rew and Wilson, 2000*; *Ruijtenberg and van den Heuvel, 2016*). Mammalian cells and cell-lines exit the cell cycle in response to various environmental changes. Quiescence, or the G0-arrest phase is one type of cell cycle arrest state, which is typically invoked in response to nutrient deprivation (*Cheung and Rando, 2013*; *Oki et al., 2014*; *Yao, 2014*). Senescence is a second pivotal type of cell cycle arrest that is often associated with aging and it is also considered as an anti-cancer mechanism (*Collado and Serrano, 2010*; *Pérez-Mancera et al., 2014*; *Sosa et al., 2014*). Tissue homeostasis requires precise and constrict control of these alternative cellular states, and impairment of these regulatory processes may result in degenerative or neoplastic diseases (*Besson et al., 2008*; *Spencer et al., 2013*; *Yao, 2014*; *Hafner et al., 2019*). The regulatory network that controls the proliferation and cell arrest, and the balance between them have been heavily investigated, yet predominantly at the transcription level since data is mostly available at the RNA level (*Bar-Joseph et al., 2008*; *Nagano et al., 2016*; *Hafner et al., 2017*; *Hernandez-Segura et al., 2017*; *Casella et al., 2019*).

mRNA translation on the other hand, specifically translation elongation, though studied extensively too (*Patil et al., 2012*; *Aviner et al., 2015*; *Rapino et al., 2018*; *Bludau and Aebersold, 2020*; *Knight et al., 2020*) remain less characterized in these systems.

tRNAs are a key molecular entity that converts the transcriptome into the proteome. The composition and abundance of the cellular tRNA pool is coordinated to match the codon demand of the transcriptome, which enables optimization of protein synthesis (*dos Reis et al., 2004*; *Gingold et al., 2012*; *Gardin et al., 2014*; *Hanson and Coller, 2017*; *Frumkin et al., 2018*). The efficiency of translation is often attributed to the mutual adaptation between supply – the abundance of each tRNA family in the cell, and the demand – the mRNA content of the transcriptome and in particular the extent of usage of each of the 61 types of codons. Efficient mRNA translation depends on supply-to-demand adaptation, and it is attained when the highly demanded codons are matched by abundantly available corresponding tRNAs (*Presnyak et al., 2015*; *Hanson and Coller, 2017*; *Rak et al., 2018*). Indeed, expression level of tRNAs was shown in recent years not to be constant and to be subject to extensive regulation that in part matches supply-to-demand (*Dittmar et al., 2006*; *Kirchner and Ignatova, 2014*; *Pan, 2018*; *Rak et al., 2018*; *Hernandez-Alias et al., 2020*). For example, cancerous cells show massive changes in expression of the tRNA pool (*Pavon-Eternod et al., 2009*; *Gingold et al., 2014*; *Zhang et al., 2018*; *Santos et al., 2019*; *Hernandez-Alias et al., 2020*). Metastatic cells too, show particular changes in their tRNA pool (*Goodarzi et al., 2016*).

By examination of diverse proliferative and non-proliferative cells, it was demonstrated that proliferating and differentiated cells induce or repress distinct sets of tRNAs, in a manner that may match the variable codon usage demand in these conditions (*Gingold et al., 2014*). In examination of diverse proliferative (normal or cancerous) cells, along with differentiated and arrested model systems, we previously defined two sets of human tRNAs, the 'proliferation-tRNAs' that are induced in proliferating cells and repressed in differentiated and arrested cells, and the 'differentiation-tRNAs' that largely manifest the opposing dynamics. Together, tRNAs from the two sets make up close to half of the human tRNA pool, the rest are tRNAs that show no consistent dynamics of expression across these conditions. This distinction between the two subsets of the tRNA pools appears to be at work also upon conversion of mouse fibroblasts into induced pluripotent stem cells (*Zviran et al., 2019*). Yet, the correlation between the expression state of the tRNA pool and the proliferative or arrested cellular state does not reveal causal effects. Are the proliferation-induced tRNAs indeed more essential during cellular proliferation than the differentiation/arrest – induced tRNAs? And which tRNAs are needed during and following the response to cell arresting signals?

To elucidate the functional essentiality of the proliferation-associated, and differentiation-associated tRNAs in diverse cellular states we mutated various human tRNA genes using CRISPR-iCas9. We succeeded to systematically CRISPR-target a significant portion of the human tRNA gene families in several human cell lines with a single sgRNA per tRNA family. This resulted in a set of cellular clones, each have a perturbed expression of one tRNA family on the background of each of the cell lines. We then assessed, in a pooled competition fashion, the relative essentiality of each tRNA family in each of these cell lines that together span a range of cellular proliferation levels. By and large, the previously defined proliferation-tRNAs were found to be more essential, on average, than the differentiation-tRNAs in most highly proliferative cell lines, and less so in slowly proliferating cells. Notwithstanding, tRNA essentiality appears to also depend on cell origin. The requirement of the various tRNAs for proper response of cells to arresting signals was found to be more complex with members from each tRNA group showing differential roles. Our results thus reveal the distinct role of various tRNAs in cellular proliferation and cell cycle arrest.

## Results

### Designing a sgRNA library that targets human tRNA gene families

In this study, we aimed at perturbing diverse human tRNA genes and to then examine the manipulation effects on various cellular phenotypes. tRNA genes appear as isodecoder gene families, that are sets of tRNA genes that share the same anticodon identity. In the human genome, tRNA families can consist of up to dozens of members, with diverse degrees of sequence similarities in the tRNA molecule outside of the anticodon itself. This situation poses a major challenge in targeting the members

of each tRNA family with CRISPR-based genomic editing. While designing the sequences of the sgRNAs to edit each tRNA gene family, we attempted to target as many as possible members of the family. In our library design we used a single sgRNA per family, choosing the one that maximizes coverage across family members (*Figure 1A*). Yet, due to sequence divergence among family members, in some families we could at best target a portion of the members, those with full or high complementarity to the sgRNA sequence (*Figure 1A and B*, *Figure 1—figure supplement 1A*). We noticed that the chosen sgRNAs, those that target most of the tRNA family members, are complementary to the functional elements along the tRNA molecule, particularly the internal promoters A and B Boxes and the anticodon (*Figure 1A and C*, *Figure 1—figure supplement 1B*). Inevitably though in many tRNA families some members had mismatches relative to their sgRNA sequence, which suggest that these tRNA genes might not be targeted by the chosen sgRNA (*Figure 1—figure supplement 1A*). Indeed, by deep-sequencing of the tRNA pool in WT HeLa cells, we found that the members that were fully matched to their respective sgRNA in each tRNA family tended to be significantly more highly expressed than those that due to sequence divergence may evade targeting (*Figure 1D*, Wilcoxon rank-sum test, $p<10^{-4}$). This result is in line with classical and more recent observations that consistently show that the rate of evolution of genes, and tRNAs in particular, correlates with expression level (*Akashi and Gojobori, 2002*; *Thornlow, 2018*). The practical and desired implication of this correlation between expression level and conservation (and hence CRISPR-based targetability in this experiment) means that although for some tRNA families we targeted only a portion of the members, we often targeted the more conserved ones, that tend to be more highly expressed, and that might hence contribute more to the functional tRNA pool.

Another challenge in applying a CRISPR-based genomic editing of tRNAs is their non-protein coding nature. When exploiting CRISPR-Cas9 based editing on protein coding genes, most of the disruptive effects result from out-of-frame Indel mutations upon repair. Yet, in non-coding genes, the effect of Indels on functionality are less trivial due to lack of a reading frame (*Ho et al., 2015*). However, the tRNA gene includes sequence elements that are critical for the functionality, such as the anticodon loop or the two internal promoter boxes. During the sgRNA library design, we attempted to direct the sgRNAs to the functional sequence elements of the tRNA (*Figure 1A and C*, *Figure 1—figure supplement 1B*), so that potentially near-by Indels would be maximally perturbing. A point in favor of our approach is that tRNAs are among the shortest RNAs in the human transcriptome, hence Indels, of even a few bases, constitute a significant portion of the molecule and may thus disrupt the secondary structure of the molecule, hence reducing functionality of the mature tRNA.

A last challenge in this sgRNA library design is the potential for off-target effects between sgRNAs and different tRNA families, or also between sgRNAs and complementary non-tRNA genes in the genome. The sgRNA design process (see Materials and methods) indeed attempts to minimize such off-targeting. A retrospective examination of the sequence complementarity between tRNA genes and the sgRNAs indeed shows that most sgRNAs fully matched only their ON-target tRNA families, while tRNA families that are considered OFF-targets have at least one mismatch (except of the sgRNA ProAGG, that fully matched several ProTGG and ProCGG genes, *Figure 1—figure supplement 1A*). Further, expanding this analysis for non-tRNA genes that could have served as OFF-targets shows that there are no such targets with fewer than two mismatches to any sgRNA (*Figure 1—figure supplement 1C*). Since editing sites located in exons are likely to have increased knockout efficiency of protein-coding gene (*Ran et al., 2013*), we focused on genes that are predicted to have Indel mutations in exons, and found only 20 such genes in combination across all 20 sgRNAs in our library (*Figure 1—figure supplement 1C*). These findings suggest that the potential for OFF-targeting in our library is limited and that hence most fitness effects due to targeting might be ascribed to ON-target tRNAs. We confirm this in expression analysis of ON and OFF targeted tRNAs in the following section.

In total we have targeted 19 out of 46 families of human tRNA genes, of which nine that constitute the 'proliferation-tRNAs', 10 that constitute the 'differentiation tRNAs' (*Gingold et al., 2014*). In addition, we targeted one pseudo-tRNA family, AsnATT.

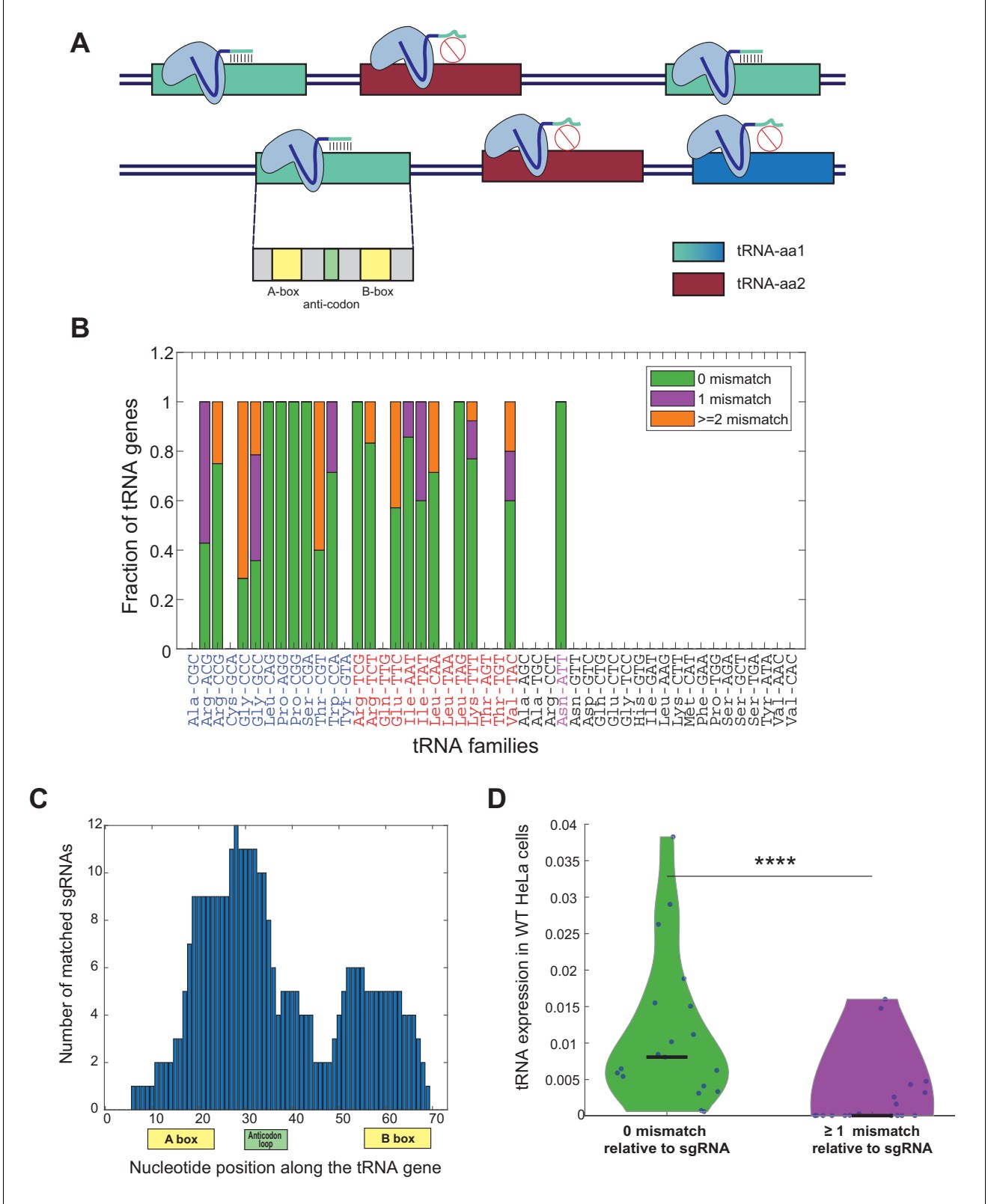

**Figure 1.** sgRNA library design for genomic editing of human tRNA genes. (**A**) A schema illustrating the sgRNA design for tRNA targeting. The hypothetical tRNA-aa1 family (blue tRNA genes) is targeted by the light-blue sgRNA. Three out of the four tRNA genes (light blue tRNA genes) are fully match to the sgRNA sequence, while the fourth gene has sequence dissimilarities (dark blue tRNA gene), thus predicted not to be targeted by the sgRNA. The hypothetical tRNA-aa2 family (bordeaux tRNA genes) is not predicted to be targeted by the light-blue sgRNA, due to lack of

*Figure 1 continued on next page*

Figure 1 continued

complementarity between the sequences. In addition, the sgRNAs are designed to target functional sequence regions along the tRNA gene to maximize the manipulation effect on the targeted tRNA. (B) A bar plot representing the sequence similarity of the tRNA genes to the corresponding sgRNAs. Each bar denotes a tRNA family in the human genome, overall, 19 out of the 46 tRNA families and one (AsnATT) out of two pseudo tRNA families were targeted. The y-axis denotes the fraction of CRISPR-targeted tRNA genes for each tRNA family (considering only tRNA genes with tRNA score >50, except of the pseudogene AsnATT tRNA family which consists of two genes with tRNA score <50 (colored in pink) [*Lowe and Chan, 2016*]). The colors in the bars describe the variety of sequence similarity of the tRNA family to the sgRNA sequence (full match in green, one mismatch in purple and two or more mismatches in orange). The tRNA identity is colored according to the differentiation (blue)/proliferation (red)/others (black) classification. (C) A histogram of the location of the sgRNA sequences along the tRNA genes. The x-axis depicts the nucleotide position along the tRNA, with the A box, B box, and the anticodon loop marked. The y-axis depicts the number of sgRNAs that are complementary to each nucleotide. (D) A violin plot describing the distribution of expression levels, in WT HeLa cells, of perfectly matched and non-perfectly matched tRNA genes to their corresponding sgRNAs. The expression of each tRNA family in HeLa cells was calculated by the sum expression of the tRNA genes, averaged over two biological repeats. The distributions are significantly different, -Wilcoxon rank-sum test, $p<10^{-4}$.

The online version of this article includes the following source data and figure supplement(s) for figure 1:

**Source data 1.** sgRNA sequences.
**Figure supplement 1.** sgRNA targeting parameters and OFF-targets potential.

## Genomic editing of proliferation-tRNAs results in negative selection and a global change of the tRNA pool in HeLa cells

The 'proliferation-tRNAs' were previously defined as tRNAs that are induced in cancer and other proliferating cells in comparison to other 'differentiation-tRNAs' that are induced in differentiated tissues and in non-dividing cells (*Gingold et al., 2014*). To test whether these two sets of tRNAs have a differential essentiality in proliferation or in response to cell arresting signals, we first set to validate that tRNA targeting by the CRISPR system results in expression perturbation of the targeted tRNAs. To examine this, we chose to CRISPR-target four tRNAs in HeLa cells - two proliferation-tRNAs, LeuTAG and ArgTCG, and two differentiation-tRNAs, ProCGG and SerCGA. Since all tRNA isodecoder genes of these four tRNA families are perfectly complementary to their respective sgRNA sequences, they are predicted to be fully targeted by the CRISPR system (*Figure 1B*, *Figure 1—figure supplement 1A*). We transduced HeLa cells expressing an inducible Cas9 (i.e. iCas9) with each of the four sgRNAs, separately and independently. Following antibiotic selection, we induced the iCas9 gene by adding Doxycycline to the cell's media for 12 days. To estimate the change in expression of the targeted tRNAs over time, we performed RNA sequencing of the mature tRNAs in each cell population in four time points along the iCas9 induction. Lastly, we calculated the fold-change in expression of the CRISPR-targeted tRNAs in treated cells relative to the WT cells (see Materials and methods).

The expression level of the CRISPR-targeted tRNAs reduced up to twofold compared to WT HeLa cells, an indication for the effectiveness of the genomic editing to reduce the expression of tRNAs (*Figure 2A*, bars marked with orange arrows). Yet, we noticed that for each CRISPR-targeted tRNA, the maximum reduction in the expression level was reached at a different day following the iCas9 induction. In particular, LeuTAG and ProCGG were maximally repressed already at day 4, while only at day 8 ArgTCG and SerCGA reached to the lowest expression level (*Figure 2A*). We then tested the expression pattern of each CRISPR-targeted tRNA throughout the iCas9 induction for each treated population. For LeuTAG, ProCGG and SerCGA, we found a similar pattern, that is a decrease in expression level, followed by a gradual recovery to basal levels (*Figure 2A*. LeuTAG day 4 vs. day 12, t-test, p=0.05; ArgTCG day 8 vs. day 12, t-test, p=ns; ProCGG day 4 vs. day 12, t-test, p=0.05; SerCGA day 8 vs. day 12, t-test, p=ns). However, even at day 12 of the iCas9 induction, the expression level of SerCGA remained low (*Figure 2A*). These differences between the tRNAs suggest that cells with CRISPR-edited SerCGA genes have a higher fitness relative to WT cells compare to CRISPR-edited LeuTAG and ProCGG genes. To test this hypothesis, for each targeted tRNA, we determined the fraction of CRISPR-edited tRNAs by computing the ratio between the number of reads corresponding to the CRISPR-edited tRNAs and the total tRNA reads, both at the genomic level (by DNA sequencing [*Figure 2—figure supplement 1A*]) and at the RNA level (from the RNA sequencing [*Figure 2—figure supplement 1B*]). At both levels, we found that for each targeted tRNA (except of ArgTCG), the dynamics of the targeted tRNA fraction along the iCas9 induction reflected the expression level. An increase in the CRISPR-edited tRNA reads in the first 4–8 days of

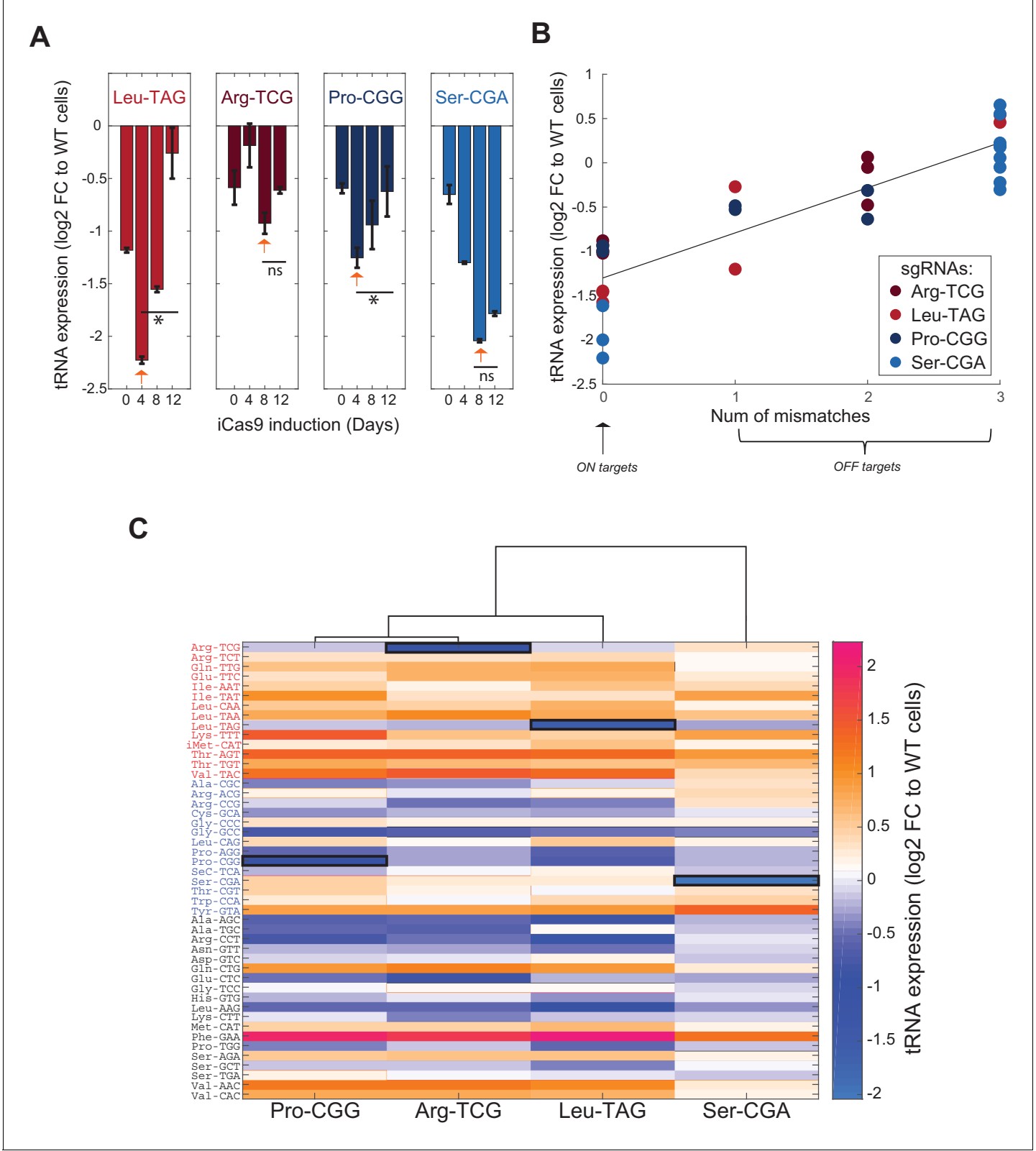

**Figure 2.** Genomic editing of proliferation-demanded tRNAs exerts a negative selection and a global change in the cellular tRNA pool in HeLa cells. (**A**) Bar plots describing the tRNA expression dynamics along the iCas9 induction for each CRISPR-targeted tRNA. The y-axis depicts the fold-change (log2) in the tRNA expression of the CRISPR-targeted tRNA in the treated cells relative to WT cells, summed over all gene members of the CRISPR-targeted tRNA family. Each bar represents a time point during the iCas9 induction (n = 2) and the error bars depicts the standard errors. The orange

*Figure 2 continued on next page*

*Figure 2 continued*

arrows point on the maximized expression reduction of the CRISPR-targeted tRNA during the iCas9 induction. The two left plots describe the proliferation edited tRNA families and the two left plots describe the differentiation edited tRNA families. * indicates p<0.05 and ns indicates non-significant p-value of two-sample T-tests comparing between the sample with the highest expression reduction and the last time point. (B) A scatter plot of the correlation between the number of mismatches of the CRISPR-targeted tRNAs (both ON-target and OFF-targets) relative to the corresponding sgRNA sequence and their change in expression in CRISPR-targeted HeLa cells (Pearson correlation, r = 0.87, p<10$^{-4}$). Each dot represents a tRNA gene family with a potential to be targeted by a sgRNA (with 0–3 mismatches), while the sgRNA is indicated by the color. The ON-targets have 0 mismatch, while OFF-targets consists between 1–3 mismatches. The y-axis depicts the fold-change (log2) in the tRNA expression of the potential targets in the CRISPR-treated HeLa cells relative to WT HeLa cells, averaged over two biological repeats. (C) A heat map representing the differential expression of the cellular tRNA pool in CRISPR-targeted tRNA cells. Each column represents a cell population with CRISPR-targeted tRNA family, and each row represents a tRNA isodecoder, grouped by their type (proliferation-red/differentiation-blue/other-black). The color code depicts the fold-change (log2) in tRNA expression in the CRISPR-targeted tRNA sample at day8 of the iCas9 induction relative to the WT sample at the same day, averaged over two biological repeats. The expression level of the ON-target tRNA in each sample is marked in black square. The dendrogram represents the hierarchical clustering of the different CRISPR-targeted tRNA samples based on changes in tRNA expression profile.

The online version of this article includes the following source data and figure supplement(s) for figure 2:

**Source data 1.** Primer sequences for genomic tRNA sequencing.

**Figure supplement 1.** Fraction of edited tRNAs in CRISPR-targeted HeLa cells from genomic and mature tRNA sequencing.

the iCas9 induction is followed by an increase of the intact tRNA reads on the expense of the CRISPR-edited tRNA reads (*Figure 2—figure supplement 1A–B*). Together, the observed dynamics suggest that most CRISPR editing occurs in the first 4 to 8 days (*Yuen et al., 2017*), and it is then followed by a competition between cells with various types and extents of editing. The cell competition results in a decline in the edited form of the CRISPR-targeted tRNAs that reflects selection against edited tRNA cells. The negative selection appears to be most pronounced for LeuTAG and ProCGG edited cells, suggesting that these tRNAs are essential in proliferative HeLa cells. Conversely, cells with CRISPR-edited SerCGA genes had only a minor disadvantage and mainly continued to propagate in the population along with their un-edited counterparts. Moreover, these results propose that ProCGG tRNA, despite its original designation as a differentiation-tRNA, might have an essential role in proliferation of HeLa cells.

We next turned to address the above-mentioned concern, some sgRNAs may affect other partially complementary tRNAs' expression. We focused on the four tRNAs, LeuTAG, ArgTCG, ProCGG and SerCGA, and their corresponding sgRNAs in HeLa cells. Comparing the sequence of each of 4 sgRNAs against each human tRNA gene allows us to define for each sgRNA it's perfectly –matching ON-target and also OFF-targets with either 1, 2, or three nucleotide mismatches. Sequencing the entire tRNA pool at the RNA level under each sgRNA allowed us to assess the effects of the three levels of -mismatches on tRNA expression. We found maximal reduction for ON-targets that gradually recovered to WT levels with extent of mismatch at OFF-targets. In particular we see a positive linear correlation between the number of mismatches and the logarithm reduction in tRNA expression (*Figure 2B*, regression model, tRNA expression as a linear function of number of mismatches, had a slope of 0.5 and intercept of −1.3; Pearson correlation, r = 0.87, p<10$^{-4}$). Compared to expression reduction of ON-targets, the reduction in expression level was already lower for OFF targets with one mismatch, and much lower, even non-existing for OFF targets with two or more mismatches.

In addition to changes in the expression levels of the CRISPR-targeted tRNAs themselves, the tRNA pool in cells may feature a more complex response following the reduction of each of the individually targeted tRNAs. Such cellular response would be reflected in expression changes of other tRNAs that were not CRISPR-targeted directly or in-directly as OFF-targets. We, therefore, monitored, in HeLa cells, the change in the expression levels of all tRNAs in each of the four individually manipulated tRNA populations throughout iCas9 induction. We found that the tRNA pool does respond to genomic editing of individual tRNAs, either by induction or repression of other tRNAs, by factors as high as 2 to 4 relative to WT HeLa cells (*Figure 2C*). Further, the tRNA pool responded similarly in the CRISPR-targeting of the three tRNAs that proved to be the most essential in the above experiment, namely ProCGG, ArgTCG, and LeuTAG (*Figure 2A*, *Figure 2—figure supplement 1A–B*). In contrast, the response to the targeting of SerCGA, which is relatively less essential in HeLa cells, was mild also at the expression of the tRNA pool level, and it resembled the tRNA pool

of WT cells (*Figure 2C*). When examining the type of tRNAs which are differentially expressed in either ProCGG, ArgTCG, and LeuTAG-targeted cells, we observed that most of the up-regulated tRNAs belong to the proliferation-tRNA group, while most of the down-regulated tRNAs belong to the differentiation-tRNA group (*Figure 2C*). The other tRNAs, those that do not belong to the proliferation or differentiation-tRNA sets, showed a mixed pattern that was characterized with either up or down expression regulation (*Figure 2C*). These results suggest that the tRNA pool in HeLa cells is responsive to the reduction of essential tRNAs. In particular, proliferation-tRNAs are preferentially up-regulated, whereas differentiation-tRNAs are mostly down-regulated following expression manipulation of essential tRNAs.

## Proliferation-tRNAs are more essential than differentiation-tRNAs for cellular growth of HeLa cells

As we validated that CRISPR-iCas9 is a suitable system to perturb the tRNA expression level, we conducted a CRISPR- targeted tRNA variants competition experiment among our designed library of 20 sgRNAs, to evaluate how reduction in tRNA levels affects the proliferation of HeLa cells. We transduced HeLa-iCas9 cells with the sgRNA library in a way in which each cell in the population expresses a single sgRNA type, and induced the iCas9 gene for 14 days (*Figure 3A*). To evaluate the growth dynamics of each of the CRISPR-targeted tRNA variants among the competing cells in the population, we deep-sequenced the genomic region encoding for the sgRNAs in five time points during the iCas9 induction. Then, we estimated the relative fitness of each CRISPR-targeted tRNA variant by calculating the fold-change of the sgRNA frequency in each time point relative to day 0 (before iCas9 induction) (*Figure 3A*, see Materials and methods). Beginning from day 7 of the competition, we observed a strong difference in relative sgRNA frequency between the CRISPR-targeted tRNA groups, with some CRISPR mutants declining in frequency by 2 orders of magnitude already at day 7, and by 3 orders of magnitude at day 14 relative to day 0 (*Figure 3B*). Many of the targeted proliferation-tRNA variants showed a sharp decline in frequency in the pooled population, while many of the targeted differentiation-tRNA variants showed relatively mild change in frequency (*Figure 3B*). This is a clear indication that on average as a group, the proliferation-tRNAs are more essential for HeLa cells. Assuming an unperturbed doubling time of about one day for this cell line, a decline in frequency of some of the proliferation-tRNAs by a factor of ~4000 compared to the total targeted population, obtained over 14 days, indicates almost complete arrest of cell doublings or cell death immediately upon iCas9 activation. CRISPR-targeting of the pseudo tRNA AsnATT, a very lowly expressed tRNA in HeLa cells (undetected in tRNA sequencing), elevated the relative frequency of the cells that carried its sgRNA, indicating very small contribution to fitness (*Figure 3B*).

We were further interested in revealing whether the expression levels of each of the tRNA in WT HeLa cells can explain the effect on their relative fitness upon CRISPR-targeting. We found a negative correlation between the expression levels of each of the CRISPR-targeted tRNAs in WT HeLa cells and the relative fitness of their CRISPR-edited tRNA variants, indicating higher essentiality of highly expressed tRNA (*Figure 3C*, regression model, relative fitness as a linear function of tRNA expression, had a slope of $-219.5$ and intercept of 0.26; Pearson correlation, r = $-0.71$, p<$10^{-3}$). This is in line with observations made in evolutionary biology showing that highly expressed genes tend to be more essential than lowly expressed ones (*Krylov et al., 2003*). We next wanted to examine if the higher essentiality of the proliferation-tRNAs could be explained by their being more highly expressed or targeted more efficiently than the differentiation-tRNAs in HeLa cells. Yet we found no significant difference in the distribution of expression level between the proliferation and differentiation-tRNAs in WT HeLa cells (*Figure 3—figure supplement 1A*, Wilcoxon rank-sum test, p=ns), or in the targeting efficiency, namely the fraction of tRNA genes with full complementarity to their respective sgRNA, between the proliferation and differentiation- tRNAs (*Figure 3—figure supplement 1B*, Wilcoxon rank-sum test, p=ns). These results indicate that the higher essentiality of the proliferation-tRNAs (*Figure 3C*, *Figure 3—figure supplement 1C*, Wilcoxon rank-sum test, p<0.05) cannot be explained by mere expression level or targeting efficiency differences between the proliferation and differentiation-tRNAs.

The relative fitness of CRISPR-targeted HeLa variants, assessed in the competition experiment, might be influenced by contributions of OFF-targeted tRNAs. We thus aimed to better approximate the real relative fitness contribution of each tRNA family from the observed fitness contribution of that family, and the extent of OFF-targeting. Assuming a linear model, in which fitness effects

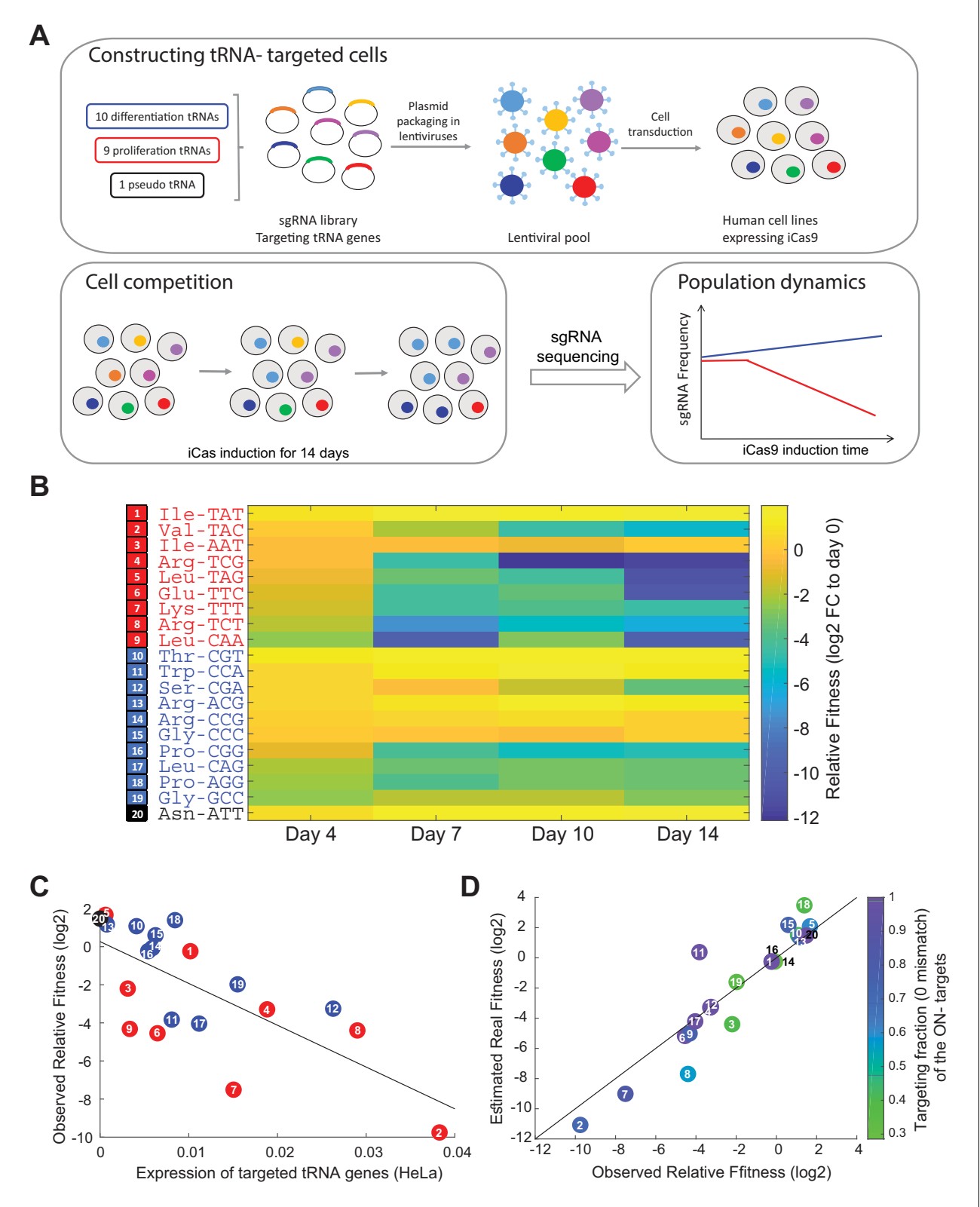

**Figure 3.** Proliferation-tRNA are essential for cellular growth in HeLa cells. (**A**) The experimental design. We designed a CRISPR-sgRNA library, in which each sgRNA targets specific tRNA gene family. Overall, we targeted nine proliferation-tRNAs, 10 differentiation- tRNAs and one pseudo tRNA family. Following cloning of the sgRNAs into a lenti-plasmid, we produced a lenti-viral pool that contained the entire sgRNA pool. Then, we transduced human cell lines (HeLa, WI38 fast and WI38 slow) expressing an inducible Cas9 with the lenti-viral sgRNA pool. We performed a CRISPR-edited tRNA cell

*Figure 3 continued on next page*

*Figure 3 continued*

competition by induction of the iCas9 in parallel to antibiotic selection (two biological repeats for each cell line). The iCas9 induction continued for 14 days, during which we sampled the heterogonous population every 3–4 days. Lastly, we deep-sequenced the sgRNAs in each sample, to evaluate the growth dynamics of the targeted-tRNA cells. (B) A heat map representation of the relative fitness of CRISPR-targeted tRNA variants in HeLa cells. Each row represents a CRISPR-targeted tRNA variant, colored according to the tRNA classification (proliferation-tRNAs in red, differentiation-tRNAs in blue and pseudo tRNA in black). Each column represents a time point during the iCas9 induction. The color code depicts a proxy of each row's relative fitness - fold-change (log2) of the sgRNA read frequency in each time point relative to the sgRNA read frequency in day 0 of the iCas9 induction (see Materials and methods). The values were averaged over two biological repeats. (C) A scatter plot of the correlation between the expression of tRNAs in WT HeLa cells and their relative fitness upon CRISPR-targeting (regression model, relative fitness as a linear function of tRNA expression, had a slope of −219.5 and intercept of 0.26; Pearson correlation, r = −0.71, p<10$^{-3}$). Expression levels for each targeted tRNA families is summed over all isodecoder genes of the family and is averaged over two biological repeats. Observed relative fitness of the CRISPR-targeted tRNA cells is shown given frequency of each tRNA family in day 7 of the competition. The colors and numbers denote the tRNA group. (D) Shown here is the estimated real relative fitness (log2) of each tRNA-targeted HeLa cells based on the observed relative fitness (log2), using a linear equation system. The observed and estimated real essentiality are correlated (regression model, estimated real relative fitness as a linear function of observed relative fitness, had a slope of 1.19 and intercept of 0.37; Pearson correlation, r = 0.93, p<10$^{-5}$). The color code depicts the fraction of isodecoder genes of each ON-target tRNA family with full complementarity to the respective sgRNA. tRNA numbering in sub figures C and D is identical to those presented in sub figure B. The online version of this article includes the following figure supplement(s) for figure 3:

**Figure supplement 1.** Comparison of tRNA expression, targeting efficiency and relative fitness between differentiation and proliferation-tRNAs.

accumulate additively and not synergistically between ON and OFF-targeted tRNAs, we could then solve the following linear system of equations:

$$\bar{O} = M * \bar{R}$$

Where $\bar{O}$ is the vector of observed fitness contribution of each of the 20 ON-targeted tRNAs, M is a squared 20 × 20 matrix whose ij-th element depicts the estimated reduction expression of family of tRNA$_i$ by the sgRNA designed against family of tRNA$_j$, and $\bar{R}$ is the estimated real fitness contribution of each tRNA family, for which we aim to solve. The M matrix elements are evaluated from sequence similarity and estimated reduction of the tRNA expression at each level of mismatch (based on *Figure 2B*). For more details, see 'Material and methods'. Solving for the vector $\bar{R}$ reveals high correlation with the observed values $\bar{O}$ (*Figure 3D*, regression model, estimated real relative fitness as a linear function of observed relative fitness, had a slope of 1.19 and intercept of 0.37; Pearson correlation, r = 0.93, p<10$^{-5}$), attesting to the quality of the sgRNA designs which indeed maximized coverage of ON-targeting of families while minimizing OFF- targeting. Nonetheless we did observe, for some tRNAs, deviations in which the observed essentiality was either over or under estimated. Yet. Since the observed relative fitness did typically not change much we remained with these observed values for the rest of the analyses presented in this work.

## The response to CRISPR-targeting of tRNAs is dependent on the cell line origin and the growth rate

We next moved to examine the essentiality of the various tRNAs in more slow-growing cell lines. We looked for at least two human cell lines of similar origin that yet manifest different growth rates. We chose two fibroblasts cell lines that were both derived from the same original fibroblast cell line, WI38, in a serial passaging process (*Milyavsky et al., 2003*). An early and late time point along the serial passaging process yielded respectively the 'WI38 slow' cell line and the 'WI38 fast' cell line, whose doubling times are around ~72 hr and ~24 hr, respectively (compared to HeLa's ~ 20 hr). We applied the sgRNA library in WI38 fast iCas9 and WI38 slow iCas9 cell lines, and performed a cell competition assay between the different tRNA-targeted variants in each of the two cell lines, as was done for HeLa cells. Note that the selection marker for sgRNA-transduced cells was Puromycin, while WI38-derived cell lines have acquired resistance to Puromycin during their original immortalization process (*Milyavsky et al., 2003*). Yet, this did not hinder our analyses since sequencing the sgRNAs enabled us to examine only the cells that did harbor the sgRNA plasmid. We deep-sequenced the genomic region encoding for the sgRNAs at different time points during the competition, and used it to estimate the relative fitness of each edited tRNA variant for each cell line at day 7 of the competition.

We examined the correlation between the relative fitness of different CRISPR-targeted tRNA variants in HeLa, WI38 fast and WI38 slow cell lines. Overall, tRNAs tended to show similar relative fitness values in all cell lines (*Figure 4* and *Figure 4—figure supplements 1–2*; Person correlation, HeLa vs WI38 slow: r = 0.54, p<0.05; WI38 fast vs WI38 slow: r = 0.81, p<10$^{-4}$; HeLa vs WI38 fast: r = 0.76, p<10$^{-4}$). Yet, a more detailed comparison revealed differences between the cell lines. Comparing tRNA essentiality in WI38 slow cells and HeLa cells, that is the slowest to the fastest cell lines in this collection, revealed a marked difference in tRNA essentiality (*Figure 4A*). While HeLa cells were more sensitive to CRISPR-targeting of most of the proliferation-tRNAs but only mildly

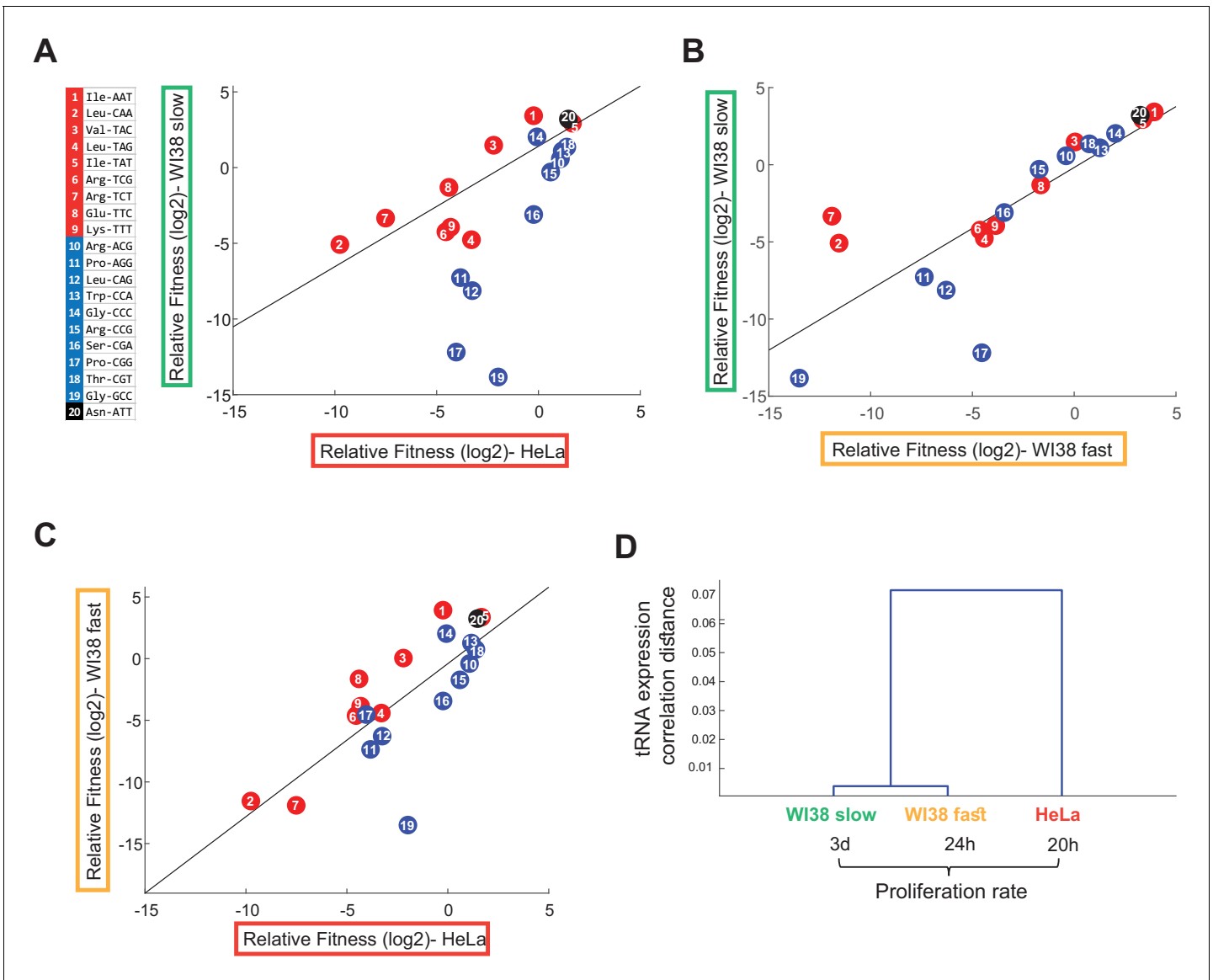

**Figure 4.** The tRNA essentiality depends on cell line origin and the proliferation rate. (**A-C**) Scatter plots that indicate the correlation between the relative fitness (log2) of the edited tRNA cells between all pairwise combinations of three cell lines (Person correlation, (**A**) HeLa vs WI38 Slow: r = 0.54, p<0.05; (**B**) WI38 Fast vs WI38 Slow: r = 0.81, p<10$^{-4}$; (**C**) HeLa vs WI38 Fast: r = 0.76, p<10$^{-4}$). The relative fitness of each CRISPR-targeted tRNA variant in each cell lines was determined based on day 7 of the cell competition, and was averaged over two biological repeats. (**D**) Hierarchical clustering of the tested cell lines based on the tRNA pool expression in WT cells.

The online version of this article includes the following figure supplement(s) for figure 4:

**Figure supplement 1.** Comparison between relative fitness of tRNA- targeted cell in different time points of iCas9 induction.
**Figure supplement 2.** Comparison between relative fitness of tRNA- targeted cell in different time points of iCas9 induction.
**Figure supplement 3.** Comparison of tRNA expression and tRNA editing between cell lines.

sensitive to differentiation-tRNA targeting, WI38 slow cells showed enhanced sensitivity to CRISPR-targeting of differentiation- tRNAs, and much lower sensitivity to targeting of proliferation-tRNAs (*Figure 4A*, regression model, proliferation-tRNA-targeted WI38 slow cells as a linear function of proliferation-tRNA- targeted HeLa cells, had a slope of 0.79 and intercept of 1.4; differentiation-tRNA-targeted WI38 slow cells as a linear function of differentiation-tRNA-targeted HeLa cells, had a slope of 2.35 and intercept of −1.82). When comparing WI38 slow to WI38 fast cells (*Figure 4B*), we observed that most tRNA targeting affected the cell's fitness similarly in the two cell lines, yet, three CRISPR-targeted tRNA variants differed in fitness (*Figure 4B*, regression model, proliferation-tRNA-targeted WI38 slow cells as a linear function of proliferation-tRNA-targeted WI38 fast cells, had a slope of 0.5 and intercept of 0.07; differentiation-tRNA-targeted WI38 slow cells as a linear function of differentiation-tRNA-targeted WI38 fast cells, had a slope of 1.1 and intercept of −0.26). Two of the three are proliferation-tRNAs that showed lower fitness in WI38 fast cells than in WI38 slow cells, and the third is a differentiation-tRNA that was more essential in the WI38 slow cells. We also observed higher sensitivity of WI38 fast cells to differentiation-tRNA-editing compared to HeLa cells, although the difference was less pronounced than the effect of differentiation-tRNA-editing in WI38 slow (*Figure 4C*, regression model, proliferation-tRNA-targeted WI38 fast cells as a linear function of proliferation-tRNA-targeted HeLa cells, had a slope of 1.5 and intercept of 2.5; differentiation-tRNA-targeted WI38 fast cells as a linear function of differentiation-tRNA-targeted HeLa cells, had a slope of 1.5 and intercept of −1.9).

To explore the possibility that the differences in essentiality of specific tRNAs detected in different cell types are due to potential differences in their expression levels in these cells, we sequenced the tRNA pool in WT cells in all cell lines. Cell line clustering based on the tRNA expression shows higher similarity between the WI38-derived cell lines than to HeLa cells (*Figure 4D*), although by and large the expression level of the different tRNA families is highly correlated between the cell lines, with correlation values ranging from 1 < R < 0.92 (*Figure 4—figure supplement 3A*). This analysis suggests that tRNA essentiality in different cell lines cannot be explained simply by differences in expression level. We further tested whether iCas9 targeting levels differ between different cell lines. For that, we sequenced the genomic regions of tRNA genes belonging to two tRNA families, Leu-TAG and SerCGA, in both WI38 fast cells and HeLa cells transduced with the corresponding sgRNAs. We found that the fraction of edited tRNA genes in WI38 fast cells was about three times lower than in HeLa cells (*Figure 4—figure supplement 3B*). This could be explained by lower efficiency of the CRISPR system in WI38 fast cells (*Lino et al., 2018*), or a higher sensitivity of these cells to genomic editing which results in loss of edited cells from the population (*Haapaniemi et al., 2018*; *Ihry et al., 2018*). Regardless to the exact reason, differences in targeting efficiency between the cell lines may affect the absolute fitness values calculated for each cell line and hence the slopes of the regression lines presented in *Figure 4*. Yet our main results, namely the ranking of essentiality of tRNAs *within* a cell line are immune from such potential caveat, and we can indeed determine changes in relative essentiality of different tRNAs in the different cell lines. These results indicate that tRNA essentiality depends both on cell origin and on the proliferation status of the cell. In particular, the more proliferative cells show a higher essentiality of the proliferation-tRNAs, while slower cell lines' fitness depend more on differentiation-tRNAs. Cellular origin appears relevant too – although the WI38 slow and fast cells differ markedly in doubling time, their essentiality profile is almost identical across most tRNAs.

## The transition from proliferative to cell cycle arrest state requires a unique set of tRNA families

Having established the differential roles of tRNAs for cellular proliferation, we turned to examine their essentiality in response to cell cycle arresting conditions. We focused on two distinct cell cycle arrest states, quiescence and senescence, which are reversible and irreversible G0 states, respectively.

To assess the role of tRNAs in entering these arrested states, we expressed the tRNA-CRISPR library described in *Figure 1* and *Figure 3A* in WI38 fast iCas9 cells. Besides the CRISPR system, we expressed in those cells a mCherry gene downstream to an endogenous promoter of human p21 (p21p-mCherry), a known marker for arrested cells. The p21 protein is a cyclin-dependent kinase inhibitor that promotes the entrance to cell cycle arrest. p21 is a primary mediator of the p53 pathway in response to DNA damage, which results in the loss of proliferation potential and induction of

senescence (*Abbas and Dutta, 2009*; *Rufini et al., 2013*). In addition, studies have shown that high p21 expression is essential for the transition to the quiescence state (*Perucca et al., 2009*; *Overton et al., 2014*).

We stably introduced a p21p-mCherry construct into WI38 fast iCas9 cells, followed by the creation of a clonal population that originated from a single cell. Then, we transduced the sgRNA library into the clonal WI38 fast iCas9 and p21p-mCherry cells. After that, we applied antibiotics selection, followed by induction of the iCas9 for 3 days to allow editing of the tRNA genes while minimizing selection based on the relative fitness of each cell in the culture. Then, we split the transduced population into three populations that were each allowed to grow continuously, yet under different conditions (see Materials and methods). Quiescence was induced by growing the cells in a serum-free medium (*Figure 5A*, 'Q' population). Senescence was induced by Etoposide at a sub-lethal concentration (*Figure 5A*, 'S' population). The untreated population continued to grow in normal conditions (*Figure 5A*, 'U' population). After 2 days under treatment, we measured the mCherry levels (normalized to the cell size, using the forward scatter (FSC) measure) of each of the three populations using a flow cytometer. We observed a significant difference in the mCherry/FSC distributions between the three populations (*Figure 5A*, Wilcoxon rank-sum test, untreated vs quiescence: $p<10^{-4}$, untreated vs senescence: $p<10^{-4}$, quiescence vs senescence: $p<10^{-4}$), indicating differences in the regulation of p21 expression in response to the different types of arrest signals. The senescent population showed the widest distribution, with the highest mCherry/FSC value (*Figure 5A*). As expected, the untreated population showed the tightest distribution, with the lowest mCherry/FSC value (*Figure 5A*). The histogram of the quiescence population is similar to that of the untreated, with a small shift towards the higher mCherry/FSC values (*Figure 5A*). Cell cycle analysis of the treated and untreated populations revealed that cells under serum starvation arrested in G1 state, were cells treated with low dosage of Etoposide arrested in G2 state (*Figure 5—figure supplement 1A*) . We further found that Etoposide- treated cells up- regulate SASP (Senescence-Associated Secretory Phenotype) genes 1 week following the Etoposidetreatment. (*Figure 5—figure supplement 1B*) These results indicate that serum starvation and low dosage of Etoposide indeed triggered the WI38 fast cells to enter into quiescence and senescence state, respectively.

We could then progress towards a multiplexed assay for the essentiality of each of the 20 CRISPR-targeted tRNAs for responding to the signal and carrying out a program for cell arrest. Since this assay did not include the cell competition phase, we were able to accurately assess the differences in sgRNA abundance between untreated and arrest-triggered cells, which infer on tRNA essentiality for entering into a non-dividing state. We reasoned that cells with CRISPR-targeted tRNAs that are important for the transition from the base condition to a non-dividing state are likely to be underrepresented in the population of responding cells, that is cells with high p21p-mCherry levels. We have thus sorted each of the three populations based on the mCherry/FSC values, and sampled cells from the top and bottom 5% of each of the respective populations (i.e. High ('H') and Low ('L') bins, respectively, *Figure 5A*). We hypothesized that the high mCherry/FSC samples will be enriched with responding cells, while the low mCherry/FSC samples will be enriched with non-responding cells (*Figure 5A*). To determine tRNA essentiality, we sequenced the genomic region of the sgRNAs in each sorted sample and calculated the sgRNA abundance (*Figure 5A*). We examined the diversity of the sgRNA read count in each sorted population of the different treatments, normalized to the sgRNA read count of the ancestor sample (the edited tRNA cell population, without any treatment). We thus obtained the normalized abundancy profile of 20 tRNAs at six samples (three conditions, times high and low mCherry level for each). To assess the similarity of tRNA abundancy profiles of the six samples we used two-way hierarchical clustering of the samples and the tRNAs (*Figure 5B*). Clustering samples across the 20 CRISPR-targeted tRNAs shows two main clusters (*Figure 5B*). One cluster consists of all high mCherry/FSC samples from all three treatments (called, U-H, Q-H and S-H), as well as senescence low mCherry/FSC sample (i.e. S-L), suggesting that in the senescence treatment, even the Low mCherry/FSC cells, S-L sample, contains responding cells. The second cluster contains the two low mCherry/FSC samples from the untreated and quiescence populations (i.e. U-L and Q-L). Upon clustering the 20 CRISPR-targeted tRNAs across the samples we could detect some tRNAs that appear essential for entry into the arrested state (*Figure 5B*). Interestingly, these tRNAs do not belong consistently to either the proliferation-tRNA or differentiation-tRNA sets. Yet, overall, same tRNAs tend to be more essential for both entries to quiescence and senescence.

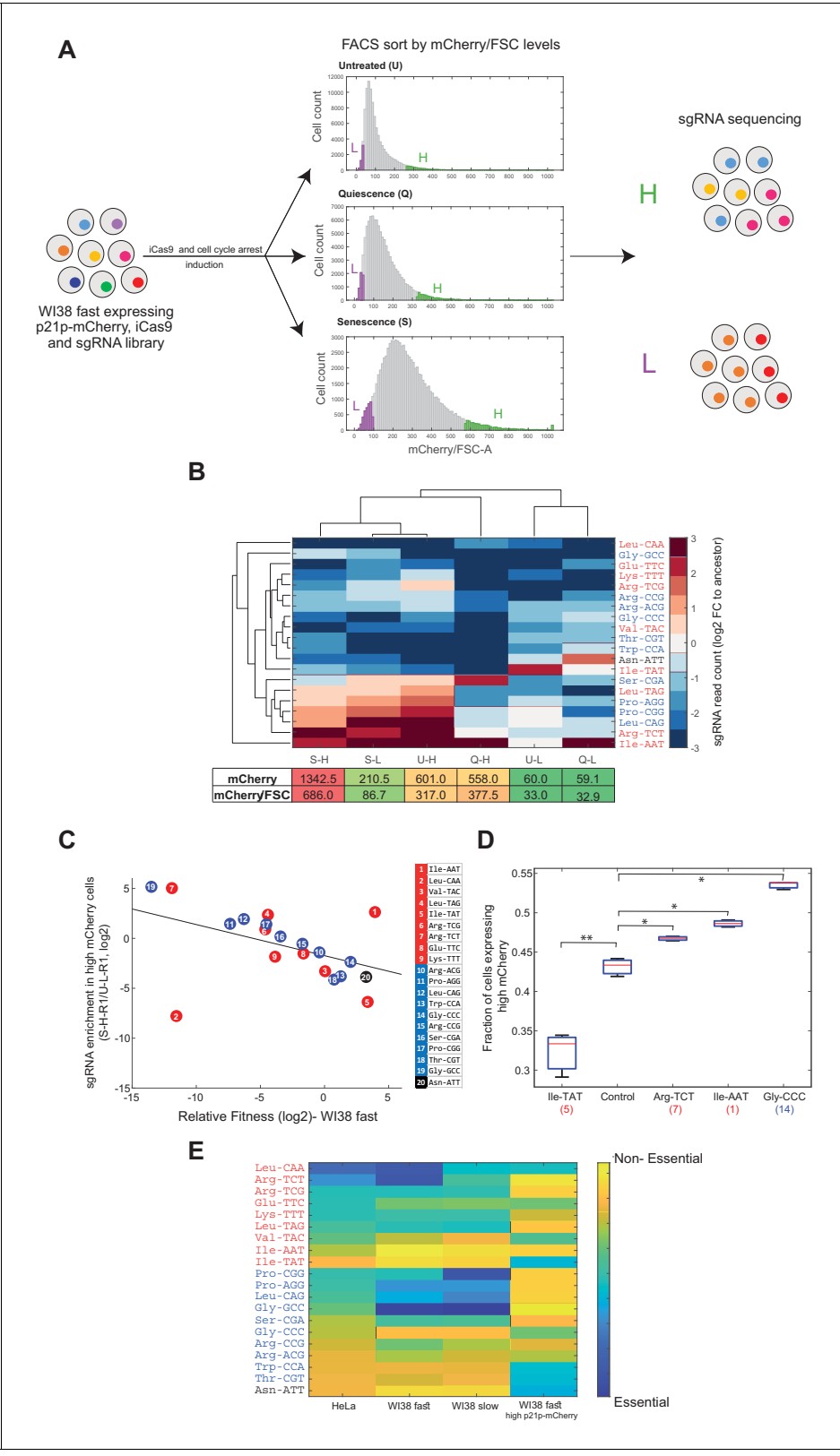

**Figure 5.** Essentiality of tRNAs for the response to cell cycle arresting signals. (**A**) A multiplexed assay for tRNA essentiality for growth arrest signals. The tRNA-sgRNA library was transduced into WI38 fast cells that express iCas9 under doxy inducible promoter and mCherry gene under the endogenous promoter of human p21. After the induction of iCas9, we split the cell population (ancestor sample) into three populations, each treated with different conditions to stimulate the entrance to a cell cycle arrest state. After 2 days, we measured the mCherry/FSC levels of each

*Figure 5 continued on next page*

*Figure 5 continued*

population using FACS. The distributions of the three populations, based on the mCherry/FSC are shown in the middle (U - untreated; Q - quiescence; S - senescence). Then, each population was sorted according to the mCherry/FSC ratio, while sampling the top and bottom 5% of each population (High bin-labeled in green; Low bin-labeled in purple). Then, we extracted the genomic DNA of each sample and deep sequenced the sgRNAs, looking for enriched and depleted sgRNAs (relative to the ancestor sample). (B) Hierarchical clustering of the sorted samples and the sgRNAs based on the averaged changes (two biological repeats) in the sgRNA read count of each sorted sample normalized to the ancestor sample (log2). The lower table depicts the mean mCherry level and the mean mCherry/FSC ratio of each sorted population, based on the FACS measurements. (C) A scatter plot comparing the essentiality of tRNAs to cellular proliferation and enterance into cell arrest in WI38 fast cells. (regression model, sgRNA abundancy in high p21-mCherry cells as a linear function of the relative fitness of tRNA-targeted WI38 fast cells, had a slope of −0.38 and intercept of −1.47; Pearson correlation, r = −0.45, p<0.05). Colors of tRNA families are as in previous Figures. (D) A box plot presenting the fraction of cells expressing high mCherry levels (>1700 relative fluorescence intensity (arbitrary units)) in each CRISPR-targeted tRNA variant (three biological repeats). The control sample is expressing a random sgRNA with no target sequence in the human genome. The number in parentheses on the x-axis denotes the number of the targeted tRNA family genes, as in subfigure C. A two-sample t-test was conducted to compare each sample to the control. * indicates p<0.05; ** indicates p<0.01. (E) A heat map summarizing the essentiality of each tRNA to different cell lines and proliferation states. Each row represents a tRNA family, classified to proliferation (red), differentiation (blue), or pseudo (black) tRNAs. Each column represents a cell line or condition. The color code depicts the essentiality of the tRNA. The essentiality is the (log2) fold-change in sgRNA read count in each experiment (as described in; *Figure 4A Figure 4B*; *Figure 4C* and subfigure C).

The online version of this article includes the following figure supplement(s) for figure 5:

**Figure supplement 1.** Serum starvation and Etoposide treatments induced quiescence and senescence respectively in WI38 fast cells.

---

We next assessed the correlation between the relative fitness of the CRISPR-targeted tRNA variants in WI38 fast cells (*Figure 4B* x-axis) and their abundance in high p21-mCherry cells. We found a negative correlation throughout all but 2 of the targeted- tRNAs (*Figure 5C*, regression model, sgRNA abundancy in high p21-mCherry cells as a linear function of the relative fitness of tRNA-targeted WI38 fast cells, had a slope of −0.38 and intercept of −1.47; Pearson correlation, r = −0.45, p<0.05). It seems that tRNAs that are essential for proliferation are relatively less essential for entering into growth arrest. Thus, although the transition of cells into an arrested state has a more complex dependency on various sub-set of the tRNA pool it appears to necessitate tRNAs that are distinct from those needed for proliferation.

CRISPR-targeting of certain tRNAs may triggers by itself, cell cycle arrest, independently of external induction. Under this alternative model, those tRNAs that appear to be most essential for growth arrest in response to the external signals actually trigger arrest to the lowest level upon their own targeting in otherwise-untreated cells. To address this possibility, we examined the distribution of p21-mCherry levels of certain CRISPR-targeted tRNA variants at the absence of any external arresting condition. We found that certain CRISPR-targeted tRNA variants (Arg-TCT, Ile-AAT and Gly-CCC) show elevated cell cycle arrest levels, as deduced from a higher fraction of high p21-mCherry levels in those mutants compared to a control sample (which carries a random sgRNA without any target site in the human genome) (*Figure 5D*. CRISPR-targeted tRNA variants vs control sample, t-test, p<0.05). On the other hand, genomic editing of Ile-TAT tRNA genes restricts the transition process, as inferred from the smaller fraction of high p21-mCherry cells compared to the control sample (*Figure 5D*. targeted Ile-TAT sample vs control sample, t-test, p<0.01). These results indicate that the editing of certain tRNAs indeed triggers, directly or indirectly, cell cycle arrest. In contrast, CRISPR-targeting of other tRNAs is crucial for the transition process from proliferative to arrested state, while growth itself is not dependent on these tRNAs.

*Figure 5E* shows a summary of tRNA essentiality across cell types and proliferation states. In general, we observed that while most of the proliferation-tRNAs are essential in proliferating cell lines (HeLa, WI38 fast, and to a lesser extent in WI38 slow), only part of the differentiation-tRNAs is essential for these cell lines. Yet, we identified tRNAs whose targeting affected differently the different proliferative cell lines. For example, ArgTCT is highly essential to HeLa and WI38 fast, while WI38 slow cells are less dependent on this tRNA family. GlyGCC is mildly important for the growth of HeLa cells, while WI38 fast and WI38 slow cells are highly dependent on this tRNA family. Overall, the tRNAs can be classified into four main groups (*Figure 5E*): tRNAs that are essential to all cell lines and proliferation states, such as LeuCAA; tRNAs that are essential to proliferative cell lines and less essential to the transition to arrested state, such as ArgTCG and ProAGG; tRNAs that are essential to transition from proliferation to arrest state, while proliferating cells are less dependent on

them, as IleTAT and ThrCGT; tRNAs that are relatively dispensable in all cell lines and proliferation states, like IleAAT. Interestingly, the pseudogene tRNA AsnATT appears dispensable in all proliferative assays but essential for the translation into cell cycle arrest.

## Discussion

In this work, we aimed to decipher the causal relation between the tRNA expression and the cellular state in human cells. For that, we manipulated the cellular tRNA pool in various cell lines and proliferation states by targeting 20 different tRNA families using CRISPR-iCas9. We found that cells from different origins and proliferation rates are dependent on distinct sets of tRNAs. Mainly, we identified the proliferation-tRNAs, as those whose CRISPR-edited are deleterious to highly proliferating cells. We found that slowly proliferating fibroblasts are more sensitive to the previously defined 'differentiation-tRNAs' (*Gingold et al., 2014*). However, here we found certain tRNAs that upon CRISPR-editing do not behave as expected by their original proliferation-differentiation classification. In particular, tRNAs IleAAT, IleTAT, and ValTAC, which were originally defined as proliferation-tRNAs are no longer assigned to that category, while ProCGG, ProAGG, and LeuCAG that were originally regarded as differentiation-tRNAs, should now be considered as proliferation-tRNAs since they are essential for cellular proliferation.

Additionally, we explored how the expression manipulation of tRNA families affects the ability of proliferative cells to respond to cell cycle arresting signals by entering into quiescence or senescence. We found that several tRNAs that are essential for growth are relatively dispensable for the entry into cell arrest, while tRNAs that are less essential for growth were more likely to be needed for mediating the arresting response to external stimuli. Indeed, we observed a negative correlation, across the 20 manipulated tRNAs, between their essentiality in growth and in cell arrest (*Figure 5C*). Yet, the actual identity of the tRNAs that are essential for cell arrest does not map precisely onto the distinction between the two tRNA sets. We note, though, that the previously defined 'proliferation-tRNAs' probably included tRNAs that serve in the translation of genes that carry out functions broader than mere cell division, for example transcription, translation, etc. These cellular functions are surely needed in additional cellular states such as when cells enter an arresting state (*Hernandez-Segura et al., 2017*; *Casella et al., 2019*). It is thus natural that the process of cell cycle arrest, and presumably also differentiation processes, not examined here, might also depend on some of these previously defined proliferation-tRNAs.

Apart from cellular proliferation rates, we found that tRNA essentiality can differ based on cell origin. Indeed, when we cluster the three cell lines studied here, HeLa, WI38 fast, and WI38 slow, based on tRNA expression we clearly see that the two WI38 lines resembles one another the most (*Figure 4D*). Also, in tRNA essentiality patterns these two related cell lines resemble one another, although at the essentiality levels some differences emerge (*Figure 5E*). It would have been interesting to examine the tRNA pool in fully cancerous tumors derived from these lines. Intriguingly, the relative expression of tRNA isoacceptors is largely similar across cell lines, as was shown here (*Figure 4—figure supplement 3A*) and in previous studies (*Dittmar et al., 2006*; *Zhang et al., 2018*). However, examination of the tRNA expression in primary cells revealed differential expression across tissues in eukaryotes (*Dittmar et al., 2006*; *Sagi et al., 2016*). It is possible that tRNA essentiality is preserved according to the tRNA expression of the origin of the cell line.

To the best of our knowledge our study provides a first demonstration of the feasibility of CRISPR-targeting of complex multi-gene families. As elaborated in the 'Results' section, designing and applying such method has several challenges, such as sequences dissimilarities between the gene copies within a family on one hand, and potential similarities across families on the other hand. Careful design of sgRNA sequences may still allow maximal targetability within families while minimizing potential OFF-targeting. Our linear algebraic model, allows under certain simplifying assumptions to approximate the real essentiality values that would have been measured unless OFF-targeting (*Figure 3D*). This model could apply in future for additional multi-gene families' knockout studies. The present sgRNA design principles can also apply for CRISPR-targeting of protein-coding gene families. Hence, insights from our sgRNA design can benefit for studying differential role of paralogs and gene duplication.

Changes in the expression of tRNAs in cancer are well established. Furthermore, there are several lines of evidence that point to causal effects of tRNAs in cancerous growth and metastasis formation

(*Felton-Edkins et al., 2003*; *Pavon-Eternod et al., 2009*; *Pavon-Eternod et al., 2013*; *Goodarzi et al., 2016*). Indeed, several studies showed the potential of tRNAs as cancer biomarkers (*Zhang et al., 2018*; *Hernandez-Alias et al., 2020*). Our current results naturally raise the possibility that tRNAs can become also new targets for therapeutic strategies. Downregulation of certain tRNAs may serve as a growth arresting treatment in the context of cancer. CRISPR-editing of specific tRNAs was found here to have a close-to-immediate halt of proliferation, even in the extreme case of HeLa cells. Crucially, our results show that some tRNAs are essential for fast growing cells, while slow growing cells, like most of the normal cells in the human body, are less depended on these tRNAs. Such selective essentiality is important as it may suggest that targeting these tRNAs in a mixture of healthy and cancerous cells may affect mainly cancer. It is worth mentioning in that respect the continuous improvement in precise delivery and activation of CRISPR technology in-vivo (*Shim et al., 2017*; *Tong et al., 2019*), such development increase the prospects of manipulation of specific tRNAs in a target cell within the body.

# Materials and methods

## Key resources table

| Reagent type (species) or resource | Designation | Source or reference | Identifiers | Additional information |
|---|---|---|---|---|
| Strain, strain background (*Escherichia coli*) | Dh5α | Thermo- Fisher | 18265017 | |
| Cell line (*Homo-sapiens*) | HeLa-iCas9 | George Church's lab (Harvard Medical school) | | |
| Cell line (*H. sapiens*) | WI38 fast | *Milyavsky et al., 2003* Moshe Oren's lab (Weizmann Institute) | | Referred to as WI-38/hTERT[fast], 484 |
| Cell line (*H. sapiens*) | WI38 slow | *Milyavsky et al., 2003* Moshe Oren's lab (Weizmann Institute) | | Referred to as WI-38/hTERT[slow], 48 PDL |
| Cell line (*H. sapiens*) | 293T | ATCC | CRL-3216 | |
| Recombinant DNA reagent | pB-Cas9 andpB-support vector | George Church's lab (Harvard Medical school) | | iCas9 plasmid |
| Recombinant DNA reagent | LentiGuide-Puro | Addgene | RRID:Addgene_52963 | sgRNA plasmid |
| Recombinant DNA reagent | PMD2.G | Addgene | RRID:Addgene_12259 | Lenti-virus packaging plasmid |
| Recombinant DNA reagent | psPAX2 | Addgene | RRID:Addgene_12260 | Lenti-virus packaging plasmid |
| Recombinant DNA reagent | p21p-mCherry-NLS | Plasmid bank of the Weizmann Institute | 2280 | |
| Sequenced-based reagent | Reverse-transcription-DNA primer | *Mordret et al., 2019* | | CACGACGCTCT TCCGATCTT |
| Sequenced-based reagent | Reverse-transcription-RNA primer | *Mordret et al., 2019* | | rArGrArUrCrG rGrArArGrAr GrCrGrUrCrGrUrG |
| Sequenced-based reagent | 3'-ligation adaptor | *Mordret et al., 2019* | | AGATCGGAA GAGCACA |
| Sequence-based reagent | *E. coli*-tyr-tRNA | Sigma-Aldrich | R3143 | |
| Sequence-based reagent | *S. cerevisia*-phe-tRNA | Sigma-Aldrich | R4018 | |

*Continued on next page*

Continued

| Reagent type (species) or resource | Designation | Source or reference | Identifiers | Additional information |
|---|---|---|---|---|
| Sequence-based reagent | sgRNAseq_F | This paper | PCR primers | GCTTACCGTAA CTTGAAAGT ATTTCGATT TCTTGG |
| Sequence-based reagent | sgRNAseq_R | This paper | PCR primers | CTTTTTCAAG TTGATAACGGA CTAGCCTT ATTTTAAC |
| Peptide, recombinant protein | TGIRT-III | InGex | 5073018 | |
| Commercial assay or kit | PEG virus precipitation kit | BioVision | K904-50/200 | |
| Commercial assay or kit | Gibson Assembly Master Mix | NEB | E2611 | |
| Commercial assay or kit | NucleoSpin miRNA kit | Macherey-Nagal | 740971.50 | |
| Commercial assay or kit | PureLink genomic DNA mini kit | Invitrogene | K182000 | |
| Commercial assay or kit | AB high-capacity cDNA reverse-transcription kit | Applied Biosystems | 4368814 | |
| Chemical compound, drug | Etoposide | Sigma-Aldrich | 33419-42-0 | 2.5 µM |
| Chemical compound, drug | Puromycin | Sigma-Aldrich | 58-58-2 | 2 µg/ml |
| Chemical compound, drug | Blasticidin | InvivoGen | BLL-38-02A | 10 µg/ml |
| Chemical compound, drug | Doxycycline | Sigma-Aldrich | 10592-13-9 | 1 µg/ml |
| Chemical compound, drug | Hygromycin | Thermo Fisher | 10687010 | 200 µg/ml |
| Software, algorithm | Kaluza | beckman | RRID:SCR_016182 | Flow cytometry analysis software |
| Software, algorithm | FlowJo software v10.2 (Tree Star) | BD biosciences | RRID:SCR_008520 | Flow cytometry analysis software |
| Software, algorithm | homerTool | *Duttke et al., 2019* | | |
| Software, algorithm | Bowtie2 | *Langmead and Salzberg, 2012* | RRID:SCR_005476 | |
| Software, algorithm | BedTools-coverage count | *Quinlan and Hall, 2010* | RRID:SCR_006646 | |
| Software, algorithm | samtools | *Li et al., 2009* | RRID:SCR_002105 | |
| Software, algorithm | Cutadapt | *Martin, 2011* | RRID:SCR_011841 | |

*Continued on next page*

*Continued*

| Reagent type (species) or resource | Designation | Source or reference | Identifiers | Additional information |
|---|---|---|---|---|
| Software, algorithm | CRISPResso | *Pinello et al., 2016* | https://github.com/pinellolab/CRISPResso2.git | |
| Software, algorithm | vsearch | *Rognes et al., 2016* | | |
| Other | Agencourt AMPure XP | Beckman Coulter | A63881 | SPRI-beads |
| Other | Hoechst 33342 | abcam | ab228551 | 10 µg/ml |
| Other | Dynabeads myOne SILANE | life Technologies | 37002D | |

## SgRNA design and cloning

sgRNA candidates targeting various tRNA families were designed by providing tRNA sequences to 'http://chopchop.cbu.uib.no/' (*Montague et al., 2014*). For each tRNA family, analyses were done for each of its unique sequences in the human genome. Then, a single sgRNA was chosen for each tRNA family that was predicted to target the maximum number of genomic copies and that had the highest sgRNA score, according to the sgRNA design tool. The list of sgRNA sequences is presented in *Figure 1—source data 1*.

For each sgRNA, the predicted off-target list was downloaded from the sgRNA design tool. The off-targets were classified to tRNAs and protein-coding genes according to the genomic location of the predicted editing site. Number of mismatches of the sgRNA relative to the off-target was taken from the sgRNA design tool output.

'Restriction-free' cloning was performed to create sgRNA plasmids for targeting tRNA families. For each sgRNA sequence, long primers were ordered and used as megaprimers. The PCR reaction was conducted using iProof master mix (X2) (Bio-Rad; 172–5310), 10 µM of Forward and Reverse primers and 50 ng of lenti-sgRNA plasmid (Addgene; 52963) in a 50 µl total volume reaction, for 12 cycles, annealing: Tm = 60℃ for 20 s, elongation: 72℃ for 10 min. To eliminate the original plasmid, PCR products were incubated with 1 µl *Dpn*I enzyme (NEB; R0176S) for 1 hr at 37℃, and then 20 min at 80℃ for inactivation. Following the *Dpn*I treatment, plasmids were transformed into DH5α competent bacteria (Thermo Fisher; 18265017) using standard heat shock transformation technique (*Sambrook and Russell, 2001*). To find recombinant plasmid, plasmids were purified Wizard Plus SV Minipreps DNA Purification (Promega; A1330) from ampicillin resistant colonies and subjected to Sanger sequencing. Primer used for the validation sequencing is TTAGGCAGGGATATTCACCA. For massive plasmid purification, NucleoBond Xtra Midi kit (Macherey-Nagel; 740412.50) was used.

## Cell culture

293 T cells (ATCC; CRL-3216) were grown in DMEM high glucose medium (Biological Industries; 01-052-1A) supplemented with 10% FBS, 1% penicillin/streptomycin (P/S) and 1% L-Glutamine.

HeLa cells (kindly given by Prof. George Church's lab) were grown in DMEM + NEAA (Life Technologies; 10938–025) - Dulbecco's Modified Eagle medium with 4.5 mg/ml D-Glucose and Non-Essential amino acids supplemented with 10% FBS, 1% penicillin/streptomycin (P/S), 1% L-Glutamine and 1% Sodium Pyruvate.

WI38 slow and WI38 Fast fibroblasts (referred to as WI-38/hTERT[slow], 48 PDL and WI-38/hTERT[fast], 484 PDL [*Milyavsky et al., 2003*]) were kindly given by Prof. Moshe Oren's lab (Weizmann Institute of Science). WI38-derived cell lines were grown in MEM-EAGLE + NEAA – Earle's salts base with Non-essential amino acids (Biological Industries; 01-025-1A), supplemented with 10% FBS, 1% penicillin/streptomycin (P/S) and 1% L-Glutamine.

The identity of all cell lines was authenticated by morphology check and growth curve analysis. In addition, all cell lines were negative to mycoplasma contamination.

For iCas9 plasmid selection, 200 µg/ml Hygromycin (Thermo Fisher; 10687010) were added to the medium and refreshed every 2 days. For sgRNA plasmid selection, 2 µg/ml Puromycin (Sigma-

Aldrich; 58-58-2) were added to the medium of HeLa cells (*Figures 2–4*). For sgRNA plasmid selection in fitness assay performed on WI38 slow and fast cell lines (*Figure 4*), 2 µg/ml Puromycin were added to the medium, although these cell lines had already Puromycin resistance acquired in their original immortalization process (the primary WI38 cells were immortalized using stable expression of pBabe-hTERT-puro [*Milyavsky et al., 2003*]). 10 µg/ml Blasticidin (InvivoGen; BLL-38-02A) was added to the medium of WI38 fast for sgRNA plasmid selection in proliferation assay (*Figure 4—figure supplement 3B*) and in cell cycle arrest treatment assay (*Figure 5*). For stable transduction, the medium containing antibiotics was refreshed every 2 days, for all cell lines. For iCas9 induction, 1 µg/ml Doxycycline (Sigma-Aldrich; 10592-13-9) was added to the medium and refreshed every 2 days.

## Generation of stable cell lines

### Generation of iCas9 carrying cell lines

HeLa cells carrying iCas9 were generously provided by Prof. George Church (Harvard Medical School).

WI38 slow and WI38 fast were seeded onto 10 cm plates such that cell confluence will be approximately 70% the next day. 5 µg of iCas9 vector pB-Cas9 and pB-support vector, kindly given by Prof. George Church lab were transfected to cells with fresh MEM-EAGLE medium (5 ml) and 15 µl of Poly-jet transfection reagent (SignaGen; SL100688). After 4 hr, the medium was replaced with a fresh MEM-EAGLE medium. Five days after the transfection, MEM medium containing 200 µg/ml Hygromycin was added to the transfected cells, refreshed every day for approximately one month, until single colonies have emerged. The cell populations that served for the following experiments originated from those single colonies.

### Generation of iCas9 cells with sgRNA plasmids

#### Step 1 - viral vector production

A 10 cm plates were covered with ~2 ml Poly-L-Lysine (Sigma; P4707) and then 293T cells (ATCC; CRL-3216) were seeded onto the covered 10 cm plates such that cell confluence will be approximately 70% the next day. A day after, 2.5 µg of PMD2.G (Addgene; 12259) and 10.3 µg psPAX2 (addgene;12260) packaging vectors were co-transfected with 7.7 µg of the appropriate sgRNA plasmids using 40 µl of jetPEI (Polyplus; 101–10N) in DMEM high glucose medium (5 ml). Note that for the pooled sgRNA experiments, the sgRNA plasmids containing the different sgRNA were added in equal amounts. After 48 and 72 hr, virus-containing medium was collected and centrifuged for 15 min at 3200 g, 4°C. Supernatant was collected to a new tube, and 1.25 ml PEG solution from PEG virus precipitation kit (BioVision; K904-50/200) was added. The virus-containing tube was stored in 4°C for at least 12 hr (over-night). Virus-contained tubes were centrifuged for 30 min at 3200 g, 4°C. Supernatant was removed and the virus pellet was suspended with 100 µl virus resuspension solution from PEG virus precipitation kit.

#### Step 2 – cell transduction

WI38 Slow, WI38 Fast and HeLa cells were seeded onto 10 cm plates cells such that cell confluence will be approximately 50% the next day. On the day of transduction (24 hr after cell seeding), medium was replaced with 5 µg/ml Poly-Brene (Sigma; TR-1003) contained medium (5 ml). Suspended viruses were added to each plate according to the calibrated titer load (MOI ~0.3).

### Generation of WI38 fast iCas9 cell lines carrying p21p-mCherry

A plasmid based on p21-mCherry-3NLS backbone (originally created by Prof. Moshe Oren's lab, purchased from the plasmid bank (#2280) of the Weizmann Institute) was used as a vector. The fragment containing human p21 promoter– mCherry-3NLS in this plasmid was replaced by the U6 region of lenti-sgRNA plasmid (Addgene; 52963) and the antibiotic resistance gene was replaced from BSD to Neo gene using Gibson Assembly cloning (Gibson Assembly Master Mix, NEB; E2611). After validation of the recombinant plasmid, it was extracted and packed into lenti-viruses. WI38 fast-iCas9 cells were transduced with the lenti-viruses as described above. After a week of G418 selection (500 µg/ml, Sigma; 108321-42-2), the cells were sorted using Flow cytometer into single cells and seeded in 96 well plate containing condition media. Flow cytometry analysis was performed

on a BD FACSAria Fusion instrument (BD Immunocytometry Systems) equipped with 488-, 405-, 561-, and 640-nm lasers, using a 100 µm nozzle, controlled by BD FACS Diva software v8.0.1 (BD Biosciences). Further analysis was performed using FlowJo software v10.2 (Tree Star). mCherry was detected by excitation at 561 nm and collection of emission using 600 LP and 610/20 BP filters. Single cells were grown for ~3 weeks until single clones appeared in the plate. The single clone that showed the highest p21-mCherry delta between arrest-treated condition and untreated condition was chosen for the following experiments.

## Measuring tRNA expression levels and editing gene variants of CRISPR-targeted tRNAs (*Figure 2*, *Figure 2—figure supplement 1*, *Figure 3C*, *Figure 3—figure supplement 1A*, *Figure 4D*, *Figure 4—figure supplement 3A-B*)

### Experimental procedure

48 hr after transduction with lentiviruses containing a single sgRNA plasmids, the cell's media was replaced with antibiotics-containing medium (2 µg/ml Puromycin) and refreshed every 2 days for 7 days in total, to allow selection of transduced cells. Then, cells were grown for additional 12 days in medium contained both 2 µg/ml Puromycin and 1 µg/ml doxycycline, to continue selection and for iCas9 induction, respectively. During the time course, the cells were diluted every 2 days in a ratio of 1:2.5. Cell samples were taken every 4 days and frozen at −80℃. In addition, cells without sgRNA (referred as WT cells) were grown with doxycycline and sampled as described above.

### Mature tRNA sequencing-library preparation and data processing

tRNA sequencing protocol was adapted from *Zheng et al., 2015* with minor modifications. Small and large RNA were extracted from frozen samples using NucleoSpin miRNA kit (Macherey-Nagel; 740971.50). 1 µg of RNA was mixed with 0.025 pmole of tRNA standards (*E. coli*-tyr-tRNA (Sigma-Aldrich; R3143) and *S.cerevisia*-phe-tRNA at ratio of 1:8 (Sigma-Aldrich; R4018)). Reverse transcription was done using TGIRT-III Enzyme (InGex; 5073018), using the primers described below. 3' adaptor was ligated to the cDNA using T4 ligase (NEB; M0202S). The cDNA was purified using Dynabeads myOne SILANE (Life Technologies; 37002D) after each step. The library was amplified using NEBNext High-Fidelity 2X PCR Master Mix (NEB; M0541S) and cleaned using SPRI-beads (Agencourt AMPure XP, Beckman Coulter; A63881), with left-side size selection protocol. Samples were pooled and sequenced using a 75 bp single read output run using MiniSeq high output reagent kit (Ilumina; FC-420–1001).

| Primer name | Primer sequence |
| --- | --- |
| Reverse-transcription-DNA | 5'-CACGACGCTCTTCCGATCTT −3' |
| Reverse-transcription-RNA | 5'-rArGrArUrCrGrGrArArGrArGrCrGrUrCrGrUrG-3' |
| 3'-ligation adaptor | 5'-AGATCGGAAGAGCACA-3' |

Read were trimmed using homerTool (*Duttke et al., 2019*). Alignment was done to the genome and mature tRNA using Bowtie2 (*Langmead and Salzberg, 2012*) with parameters – `very-sensitive-local`. Read aligned with equal alignment score to the genome and mature tRNA were annotated as mature tRNA. Reads aligned to multiple tRNA genes were randomly assigned when mapping to identical anticodon, and discarded from the analysis if aligned to different anticodon. Read count was done using BedTools-coverage count (*Quinlan and Hall, 2010*). Variant calling for detection of mutation and indels was done using samtools (*Li et al., 2009*) command 'mpileup' with the parameters: '-A -q1 -d100000'.

The fold-change in tRNA expression for each tRNA isoacceptor was calculated as follows: each tRNA read count was normalized to the total reads per sample. Then, the normalized read counts for each tRNA isoacceptor were summed up over all tRNA genes of the tRNA family in each sample (treated and WT cells) and in each time point along the iCas9 induction. Then, a ratio of the summed read count between the treated and WT samples was calculated for each time point. This procedure was done for each of the two biological repeats which were then averaged. The fraction of edited tRNA reads was calculated as follow: for each tRNA isoacceptor, the number of aligned reads with

Indel mutations around the editing site were summed up for all tRNA genes within an isoacceptor family, and normalized to the sum of maximum coverage for all tRNA genes in the vicinity of the Indel mutations. Then, a ratio of the fraction of edited reads between the treated and WT samples was calculated for each time point.

## Genomic tRNA sequencing – library preparation and data processing

Genomic DNA was purified from frozen cell samples of the single sgRNA transduced HeLa and WI38 fast cells (four sgRNAs and no sgRNA sample in each of the four time points) using PureLink genomic DNA mini kit (Invitrogene; K182000) and used as template for PCR to amplify specifically the tRNA isodecoder genes of the CRISPR-targeted tRNAs in the population. PCR reaction was conducted using KAPA HiFi HotStart ReadyMixPCR (X2) (kapabiosystems; KK2601), 10 µM of each primer and 100 ng of genomic DNA in a 60 µl total volume reaction, for 25 cycles. Primers and annealing temperatures are listed (*Figure 2—source data 1*). After PCR validation using agarose gel, the PCR product was purified with SPRI-beads using left-side size selection protocol in which the PCR product and beads were mixed at 1:1.5 ratio. Then, each sample was barcoded with a different Illumina barcode while the Forward primer was fixed and the reverse primer included the different barcodes. The barcoding PCR reaction was conducted using KAPA HiFi HotStart ReadyMixPCR Kit (X2), 10 µM of Illumina primers and 0.35–0.7 ng of PCR product in a 10 µl total volume reaction, for 15 cycles. Additional clean-up step was performed using SPRI-beads with left-side size selection protocol in which the PCR product and beads were mixed at 1:1.5 ratio. Samples were pooled in equal amounts and sequenced using a 75 bp single read output run using MiniSeq high output reagent kit (Ilumina; FC-420–1001).

After reads were trimmed using Cutadapt (*Martin, 2011*), we used CRISPResso (*Pinello et al., 2016*) to quantify NHEJ events in the CRISPR-iCas9 targeted samples. The input for the CRISPResso pipeline included both the amplicon sequences of the unedited (WT) tRNA genes and the sgRNA sequences. The minimum identity score for the alignment was set to 0. For each sample, we quantified the read count with WT amplicon or amplicons with NHEJ events.

The fraction of edited tRNA reads was calculated as follows: for each tRNA isoacceptor, the number of edited reads were summed up for all tRNA genes within the family, and normalized to the sum of aligned reads (edited and WT reads). Then, a ratio of the fraction of edited reads between the treated and WT samples was calculated for each time point.

## Evaluating the relative fitness of tRNA-targeted variants – pooled sgRNAs (*Figures 3–4*)

### Experimental procedure

48 hr after transduction with lentiviruses containing the pooled sgRNA plasmids, the cell's media was replaced with antibiotics-containing medium, to allow selection of infected cells (2 µg/ml Puromycin). Following 24 hr of selection, cells were grown for 14 days in medium contained both 2 µg/ml Puromycin and 1 µg/ml doxycycline, for selection and iCas9 induction, respectively. During the time course, cells were diluted every 2 days in a ratio of 1:2.5. Cell samples were taken every 3 or 4 days and frozen at −80°C.

### sgRNA sequencing-library preparation and data processing

Genomic DNA was purified from frozen cells samples of the sgRNA competition experiment using PureLink genomic DNA mini kit and used as templates for PCR to amplify specifically the sgRNAs in the population. PCR reaction was conducted using iProof master mix (X2), 10 µM of each primer and 20 ng of genomic DNA in a 50 µl total volume reaction, for 26 cycles, Tm = 64°C. The primers used to amplify the sgRNA region were:

> Forward primer - GCTTACCGTAACTTGAAAGTATTTCGATTTCTTGG
> Reverse primer - CTTTTTCAAGTTGATAACGGACTAGCCTTATTTTAAC

After PCR clean-up using Wizard SV Gel and PCR Clean-Up (Promega; A9281), samples were run in 2% agarose gel to ensure that the PCR product is composed of a single amplicon in the appropriate size. Next, Hiseq libraries were prepared using the sequencing library module from *Blecher-Gonen et al., 2013*. Briefly, blunt ends were repaired, Adenine bases were added to the 3' end of

the fragments, barcode adapters containing a T overhang were ligated, and finally the adapted fragments were amplified. The process was repeated for each sample with a different Illumina DNA barcode for multiplexing, and then all samples were pooled in equal amounts and sequenced. We performed a 125 bp paired end high output run on HiSeq 2500 PE Cluster Kit v4. Base calling was performed by RTA v. 1.18.64, and de-multiplexing was carried out with Casava v. 1.8.2, outputting results in FASTQ format.

De-multiplexed data was received in the form of FASTQ files split into samples. First, SeqPrep (RRID:SCR_013004) was used to merge paired reads into a single contig, to increase sequence fidelity over regions of dual coverage. The size of each contig was then compared to the amplicon's length. Next, the forward and reverse primers were found on each contig (allowing for two mismatches) and trimmed out. This step was performed for both the forward and reverse complement sequences of the contig, to account for non-directional ligation of the adaptors during library preparation. After the primers were trimmed, the contig was tested again for its length to ensure no Indels had occurred. Contigs were then compared sequentially to all sgRNAs, comparing the sequence of each contig to the sequence of each sgRNA. Any contig without a matching sgRNA within two mismatches or less was discarded. Contigs with more than a single matching sgRNA with the same reliability were also discarded due to ambiguity. Each contig that passed these filters was counted in a keyvalue data structure, storing all sgRNA types and their frequency in each sample.

The relative fitness of each CRISPR-targeted tRNA variant was estimated by calculating the fold-change of the sgRNA frequency in each time point relative to day 0 (before adding Doxycycline). We chose to explore the relative fitness of the tRNA-targeted variants based on relative frequency of their sgRNAs at day 7 (relative to day 0), due to the dynamics of the cell population composition along the iCas9 induction. In the early days of the induction, the iCas9 activity does not reach saturation (*Yuen et al., 2017*), thus the cell population is dominated by partially CRISPR-edited cells. In the late days of the iCas9 induction, the less fit CRISPR-targeted tRNA cells might be eliminated from the population due to a negative selection, a process that can result in lower frequency of the tRNA-edited cells. Nonetheless, the relative fitness estimated by the different days of the competition is highly correlated (*Figure 4—figure supplement 1A-B*, *2C*).

In an attempt to account for OFF-targeting of tRNAs by sgRNAs, we solve the system of linear equations so as to retrieve estimated 'real' relative fitness values, R, from the sequencing – based observed relative fitness values, O, given the estimated expression reduction matrix M of targeted tRNAs as a result of CRISPR targeting by each sgRNA:

$$\bar{O} = M * \bar{R}$$

The entries of the estimated expression reduction matrix M are computed as the weighted targeting efficiency of each targeted tRNA family per sgRNA. The weighted targeting efficiency of $tRNA_i$ by $sgRNA_j$ is calculated as follows:

$$coeff_{ij} = \sum p_s * f_{ijs}$$

Where $p_s$ is expression penalty denoting relative expression level reduction at any level of mismatch level s, where s is either 0, 1, 2, or 3 mismatches. The respective penalties, as deduced from mean expression reductions at each mismatch level (*Figure 2B*) are 1, 0.5, 0.2, and 0. These penalties are further multiplied by the fraction of $tRNA_i$ family that has s mismatches relative to sgRNA against $tRNA_j$ family. Summation is done over all types of mismatches per tRNA family and per sgRNA.

The underlying assumptions and constraints of this model are:

- Non-epistatic interaction between the fitness effect of targeting various tRNA families, additive contribution to fitness of ON and OFF targeted tRNAs
- A linear relationship between the fitness reduction and tRNA expression reduction upon CRISPR editing.
- The M matrix was made for only 20 tRNA families that were targeted here.
- The expression reduction penalties as a function of extent of mismatch were estimated based on a sample of the tRNAs.

### Evaluating the essentiality of edited-tRNA variants for entering into cell cycle arrest state – (*Figure 5*)

#### Experimental procedure

Clonal WI38 fast iCas9 + p21p-mCherry cells were transduced with the sgRNA library as described above. After 6 days of antibiotics selection (10 μg/ml Blasticidin), the iCas9 was induced using 1 μg/ml doxycycline for 3 days (ancestor sample). Then, the cells were split into three populations. One population continued to grow in normal conditions (untreated sample). The second population was grown with serum-free media - 0% FCS (quiescence sample). The third population was grown with media containing 2.5 μM Etoposide (Sigma-Aldrich; 33419-42-0) (senescence sample). After 2 days of treatment, the cells were sorted using Flow cytometer according to the mCherry/FSC levels, while sorting the top and bottom 5% of the population. FACS parameters are identical to that described in 'stable cell line generation' section.

#### sgRNA sequencing – library preparation and data processing

From each sorted population, as well as ancestor sample, genomic DNA was extract using PureLink genomic DNA mini kit and used as templates for PCR to amplify specifically the sgRNAs in the population. PCR reaction was conducted using 2X KAPA HiFi HotStart ReadyMix, 10 μM of each primer and 20 ng of genomic DNA extracted from the samples in a 50 μl total volume reaction, for 20 cycles, Tm = 58°C. We used shifted primers to increase library complexity. PCR products were purified with SPRI-beads using left-side size selection protocol in which the PCR product and beads were mixed at 1:1.5 ratio. The barcoding PCR and final PCR clean-up was done as described for the genomic tRNA library preparation. Samples were pooled and sequenced using a 75 bp single read output run using MiniSeq high output reagent kit (Ilumina; FC-420–1001).

Reads were trimmed using Cutadapt and then clustered into unique sequences using vsearch (*Rognes et al., 2016*). Each unique read was then aligned to the matched sgRNA sequence, allowing up to two mismatches. Finally, we stored all sgRNA types and their frequency in each sample.

### Cell-cycle analysis (*Figure 5—figure supplement 1A*)

WI38 fast cells treated with arrest-triggering conditions (described above) were incubated with 10 ug/ml of Hoechst 33342 (abcam; ab228551) for 45 min at 37°C for DNA staining. The cells were analyzed by FACS as described in 'stable cell line generation'. Cell cycle analysis was done using Kaluza software (RRID:SCR_016182).

### Evaluating the essentiality of edited tRNA variants for entering into cell cycle arrest state – single sgRNAs (*Figure 5D*)

Clonal WI38 fast iCas9 + p21p-mCherry cells were transduced with single sgRNAs as described above. After 6 days of antibiotics selection (10 μg/ml Blasticidin), the iCas9 was induced using 1 μg/ml doxycycline for 3 days. Then, the mCherry levels of each cell line were analyzed using Attune Flow Cytometer and the analysis was done using Kaluza software.

### qPCR of SASP molecules (*Figure 5—figure supplement 1B*)

Relative mRNA expression of SASP genes was measured in tRNA-targeted WI38 fast iCas9 + p21p-mCherry cell line exposed to three different treatments: untreated cells; cells after 48 hr with 2.5 μM Etoposide; cells after one week post Etoposide treatment. First, total RNA was extracted from the samples using TriZol. DNAse (Invitrogen; AM1907)-treated RNA was used as template for cDNA preparation, using AB high-capacity cDNA reverse-transcription kit (Applied Biosystems; 4368814). mRNA expression of SASP genes was then determined using quantitative RT-PCR with light cycler 480 SYBR I master kit (Roche Applied Science) and the LightCycler 480 system (Roche Applied Science), according to the manufacturer's instructions.

## Acknowledgements

We thank the Israel Science Foundation and the European Research Council for grant support. We wish to thank Tammy Biniashvili and Omer Asraf for the help with the deep-sequencing analyses. We thank Dr. Tomer Meir Salame from The Weizmann Institute of Science's Flow Cytometry Core Facility

for the help with the Flow Cytometry experiments. We thank Lior Roitman and Dr. Hilah Gal from Prof. Valery Krizhanovsky group from the Weizmann Institute of Science for the help and guidance with the cell cycle arrest experiments. Special thanks for Prof. George Church (Harvard Medical School) and Prof. Moshe Oren (Weizmann Institute of Science) for their kind contributions of cell lines and plasmids. A special thank for Dr. Hila Gingold and the Pilpel lab, for the stimulating discussions.

## Additional information

### Funding

| Funder | Grant reference number | Author |
|---|---|---|
| Israel Science Foundation | 1332/14 | Yitzhak Pilpel |
| European Research Council | 616622 | Yitzhak Pilpel |

The funders had no role in study design, data collection and interpretation, or the decision to submit the work for publication.

### Author contributions

Noa Aharon-Hefetz, Conceptualization, Resources, Data curation, Software, Formal analysis, Investigation, Visualization, Writing - original draft, Project administration; Idan Frumkin, Conceptualization, Formal analysis, Investigation, Writing - review and editing; Yoav Mayshar, Investigation, Methodology; Orna Dahan, Conceptualization, Supervision, Writing - review and editing; Yitzhak Pilpel, Conceptualization, Formal analysis, Supervision, Funding acquisition, Writing - review and editing; Roni Rak, Conceptualization, Data curation, Supervision, Validation, Project administration, Writing - review and editing

### Author ORCIDs

Orna Dahan (iD) https://orcid.org/0000-0002-8096-5085
Yitzhak Pilpel (iD) https://orcid.org/0000-0003-3200-9344

### Decision letter and Author response

Decision letter https://doi.org/10.7554/eLife.58461.sa1
Author response https://doi.org/10.7554/eLife.58461.sa2

## Additional files

### Supplementary files

• Transparent reporting form

### Data availability

Source data files have been provided for Figures 1 and 2. Sequencing data are available in GEO under the accession code GSE163611.

The following dataset was generated:

| Author(s) | Year | Dataset title | Dataset URL | Database and Identifier |
|---|---|---|---|---|
| Aharon-Hefetz N, Frumkin I, Mayshar Y, Dahan O, Pilpel Y, Rak R | 2020 | Manipulation of the human tRNA pool reveals distinct tRNA sets that act in cellular proliferation or cell cycle arrest. | https://www.ncbi.nlm.nih.gov/geo/query/acc.cgi?acc=GSE163611 | NCBI Gene Expression Omnibus, GSE163611 |

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
