## [Decision Letter]

**Acceptance summary:**

This article reports on pioneering work wherein the effects of systematically knocking out tRNA genes are directly studied. This constitutes an important milestone when considering the abundance and variability of isodecoder species and the homology between isoacceptors. This approach revealed distinctions in essentiality of tRNAs in fast vs. slow proliferating cells. The authors also investigated tRNA essentiality in senescent and quiescent cells which revealed more complex patterns. Overall, it was thought that this study is of broad potential interest inasmuch as it suggests that tRNAs have distinct essentiality in different cell types and across distinct proliferative states.

**Decision letter after peer review:**

Thank you for submitting your article "Manipulation of the human tRNA pool reveals distinct tRNA sets that act in cellular proliferation or cell cycle arrest" for consideration by *eLife*. Your article has been reviewed by three peer reviewers, including Ivan Topisirovic as the Reviewing Editor and Reviewer #1, and the evaluation has been overseen by Philip Cole as the Senior Editor. The following individual involved in review of your submission has agreed to reveal their identity: Luis Serrano (Reviewer #2).

The reviewers have discussed the reviews with one another and the Reviewing Editor has drafted this decision to help you prepare a revised submission.

As the editors have judged that your manuscript is of interest, but as described below that a number of additional experiments are required before it is published, we would like to draw your attention to changes in our revision policy that we have made in response to COVID-19 (https://elifesciences.org/articles/57162). First, because many researchers have temporarily lost access to the labs, we will give authors as much time as they need to submit revised manuscripts. We are also offering, if you choose, to post the manuscript to bioRxiv (if it is not already there) along with this decision letter and a formal designation that the manuscript is "in revision at *eLife*". Please let us know if you would like to pursue this option. (If your work is more suitable for medRxiv, you will need to post the preprint yourself, as the mechanisms for us to do so are still in development.)

Summary:

Based on their previous work showing that cell proliferation and differentiation are associated with distinct tRNA programs and codon usages, the authors employed a CRISPR-Cas9 based approach to deplete families of tRNAs belonging to "proliferation" and "differentiation" groups and test the effects of such manipulation on the fitness of cells in different proliferative states. Using competition assays, the authors provide evidence that "proliferative" tRNAs are more essential in fast-proliferating cells, while "differentiation" tRNAs exert higher essentiality in slower proliferating cells. The authors also determined the essentiality of investigated tRNAs in senescent and quiescent cells which revealed more complex patterns. Overall, it was thought that this study is of broad potential interest inasmuch as it suggests that tRNAs have distinct essentiality in different cells and across distinct proliferative states. Moreover, it was found that this constitutes pioneering work wherein the effects of systematically knocking out tRNA genes are directly studied, an important milestone by itself when considering the abundance and variability of isodecoder species and the homology between isoacceptors. Notwithstanding the overall enthusiasm for the potential importance of the study and uniqueness of the approach, it was found that several major issues should be addressed to corroborate author's conclusions as outlined below.

Essential revisions:

1) It was thought that a number of important controls were missing. The potential off-target effects of CRISPR-Cas9 method need further validation. Figure S1B should be extended in order to clarify which sgRNAs are potentially off-targeting which tRNA. The manuscript would also benefit from experimentally testing the off-target effects of some of the sgRNAs, especially those binding to other tRNA families. To accurately compare HeLa cells with fibroblasts, the authors should determine potential tRNA expression and codon usage differences between them. Moreover, the efficacy of tRNA depletion between the cell lines should be assessed. Figure 5-additional controls should be provided to ascertain that cells are indeed in quiescent and senescent states. In Figure 5A, it should be explained why the 3 day time point was used when in the most of the study it is shown that the strongest effects occur after 7 days of induction.

2) Some experimental conditions remain unclear. For instance, it is noted that sgRNA plasmids were selected by puromycin, whereby WI38 cells appear to already be puromycin resistant. It is also not clear how were competition assays carried out in cell arrested states. In general, it was thought that the authors should be more specific regarding their read-outs (i.e. specify whether proliferation or survival were monitored).

3) Several issues were raised apropos statistical analyses. In Figure 3C and D, to assess whether tested variables are truly independent, the authors should use a linear regression modelling Relative fitness ~ tRNA expression (in C) and Relative fitness ~ fraction CRISPR targeted tRNAs (in D). In addition, it is not clear why is z-transformation applied in Figure 5E? The heatmap summarizes tRNA essentiality, which in Figure 3 and Figure 5C, is depicted using an untransformed log2FC. Using z-transformed and untransformed values to estimate the same effects was thought not to be advisable. Finally, the authors should also include the number of biological replicates, types of statistical tests and their outcomes in each figure where applicable, as in some cases these are missing.

4) Several statements were found not to be adequately supported by the data. For example, the statement: "our results show that some tRNAs are essential specifically for cancerous cells and not in differentiated cells.… (and the next sentence)", was found not to be supported by the presented data. To this end, the authors are advised either to provide data corroborating these conclusions or to tone down their statements. Also, in Discussion section, given that this work is the first in systematically knocking out tRNA gene families, some comment on the potential and limitations of the method appears to be warranted.

---

## [Author Response]

Essential revisions:1) It was thought that a number of important controls were missing. The potential off-target effects of CRISPR-Cas9 method need further validation. Figure S1B should be extended in order to clarify which sgRNAs are potentially off-targeting which tRNA.

The OFF-targeting of tRNAs by the CRISPR system is a very important issue and we thank the referees for raising it here. We have taken the challenge rigorously to fully scrutinize potential OFF- targeting in our experiment, to quantify its potential effect, and to computationally clean the data from it. This is described in subsection “Designing a sgRNA library that targets human tRNA gene families”, subsection “Genomic editing of proliferation tRNAs results in negative selection and a global change of the tRNA pool in HeLa cells”, subsection “Proliferation tRNAs are more essential than differentiation tRNAs for cellular growth of HeLa cells” and in new Figure 2B and Figure 3D. In summary, our new results show that OFF- targeting has probably played a relatively minor effect and it distorted to a small extent our original measurements. In short, we performed the following: we have measured tRNA expression reduction due to various levels of mismatches between sgRNAs and OFF- targeted tRNAs (Figure 2B). We have then incorporated the resultant estimated effects of OFF- targeting on tRNA expression into a system of equations (subsection “Proliferation tRNAs are more essential than differentiation tRNAs for cellular growth of HeLa cells”) that then computes the true essentiality value of each tRNA by eliminating estimated effects from off targeted genes. We found that our original estimations were accurate (Figure 3D), and we ascribe this new result to an optimal design of sgRNA that maximized coverage of each gene family while minimizing OFF- targeting. We discuss this new analysis as means that is general enough so as to provide means to assess effects of OFF- targeting in other CRISPR experiments (Discussion section).

We changed Figure 1—figure supplement 1A's format (now presented as a "Rose Chart") that specifies, for each sgRNA, all the potentially targeted tRNA families, which are those tRNA genes that have between 0 to 3 mismatches relative to a given sgRNA sequence.

In addition, we tested protein- coding genes that can potentially be OFF- targets of the tRNAs' sgRNAs. We discussed this part in subsection “Designing a sgRNA library that targets human tRNA gene families” and Figure 1—figure supplement 1C.

2) The manuscript would also benefit from experimentally testing the off-target effects of some of the sgRNAs, especially those binding to other tRNA families.

See answer above, and in particular, we added Figure 2B which illustrates the expression level of tRNAs that are targeted by 4 different sgRNAs, each tRNA family with either 0, 1, 2 or 3 mismatches relative to their respective sgRNA sequence. We found a positive correlation between the number of mismatches and the expression level, indicating that as the OFF- targeted tRNA gene is more similar to an sgRNA of another tRNA, the OFF- target effect on expression level intensifies, thus resulting in lower expression of the targeted tRNA.

3) To accurately compare HeLa cells with fibroblasts, the authors should determine potential tRNA expression and codon usage differences between them. Moreover, the efficacy of tRNA depletion between the cell lines should be assessed.

The reviewers pointed to an important issue, which we have now addressed in full. In particular, we assessed the differences in both tRNA expression and CRISPR activity between HeLa and WI38 fibroblasts. This part is discussed in subsection “The Response to CRISPR- targeting of tRNAs is dependent on the cell line origin and the growth rate” and Figure 4D, Figure 4—figure supplement 4A and Figure 4—figure supplement 4B.

tRNA expression measurements revealed that the tRNA pool of the various cell lines we used is highly similar in expression (Figure 4D and Figure 4—figure supplement 4A). Particularly, we found that the tRNA pool of WI38 fast and WI38 slow lines was essentially identical and they were both highly similar to the HeLa's tRNA pool.

Assessment of the editing fraction of two targeted tRNA families (LeuTAG and SerCGA) in WI38 fast cells showed that the fraction of edited tRNA genes in WI38 fast cells is about 3 times lower than in HeLa cells. This could be explained by lower efficiency of the CRISPR system in WI38 fast cells, or a higher sensitivity of these cells to genomic editing which results in loss of edited cells from the population.

Importantly for us, regardless to the exact reason for such differences, they only affect the absolute fitness values calculated for each tRNA in each cell line and hence the slopes of the regression lines presented in Figure 4, but *not* the ranking between tRNAs within a cell line and hence the residuals from the regression line. Thus, the differences in essentiality between tRNAs in each cell lines will not change due to the detected difference in targeting efficiency. We concluded that based on these controls, we were able to achieve a valid comparison between the relative fitness of the different tRNA- targeted variants in HeLa and cell derivatives of WI38 fibroblasts. This analysis is now reported and discussed in subsection “The Response to CRISPR- targeting of tRNAs is dependent on the cell line origin and the growth rate”.

4) Figure 5-additional controls should be provided to ascertain that cells are indeed in quiescent and senescent states. In Figure 5A, it should be explained why the 3 day time point was used when in the most of the study it is shown that the strongest effects occur after 7 days of induction.

We agree that this is a needed validation. We thus performed now two assays, to ensure that the treated cells are indeed entering in arrest state. First, we performed cell cycle analysis to evaluate the distribution of each population along the cell cycle phases, and found that serum starvation arrests the cells in G1- typical to quiescent cells, while 48 hours with low dosage of Etoposide arrests the cells in G2, again as expected for this treatment (Figure 5—figure supplement 1A).

To assess senescence in the etoposide- treated cells, we performed qPCR to measure relative mRNA expression of two known senescence markers, Il6 and Il8. We found that after 1 week from Etoposide treatment cells upregulated Il6 and Il8 60 and 25-fold relative to untreated cells, respectively. This result indicates that these cells are indeed senescent. However, 48 hours after the Etoposide treatment to the cells, Il6 and Il8 are not yet upregulated.

We discuss this in subsection “The transition from proliferative to cell cycle arrest state requires a unique set of tRNA Families” and Figure 5—figure supplement 1.

5) Some experimental conditions remain unclear. For instance, it is noted that sgRNA plasmids were selected by puromycin, whereby WI38 cells appear to already be puromycin resistant.

The selection marker of the sgRNA plasmids was indeed an issue which we now better discuss. Indeed, the WI38 slow and WI38 fast cells were resistant to puromycin. However, sequencing the sgRNAs ensures that when we estimate the relative fitness of CRISPR-targeted tRNAs, we consider only cells that harbor the sgRNA, which constitute the only relevant subpopulations. This holds for both HeLa and WI38- derived cell lines. We are thus certain that this does not compromise any inference made here.

We refer to this issue in subsection “The Response to CRISPR- targeting of tRNAs is dependent on the cell line origin and the growth rate”, and in the Materials and methods section.

6) It is also not clear how were competition assays carried out in cell arrested states.

Our answer here is the same as for point #6 above, and we further emphasize that in the cell arrest experiments, we didn’t perform a cell competition assay, but rather we examined which tRNA- targeted variants are more represented in different mCherry levels. We clarified this issue in subsection “The transition from proliferative to cell cycle arrest state requires a unique set of tRNA Families”.

7) In general, it was thought that the authors should be more specific regarding their read-outs (i.e. specify whether proliferation or survival were monitored).

We rephrased the text to make the read-outs clearer in several point along the Results section:

– Subsection “Genomic editing of proliferation tRNAs results in negative selection and a global change of the tRNA pool in HeLa cells”:

“To test whether these two sets of tRNAs have a differential essentiality in proliferation or in response to cell arresting signals, we first set to validate that tRNA targeting by the CRISPR system results in expression perturbation of the targeted tRNAs.”

– Subsection “Proliferation tRNAs are more essential than differentiation tRNAs for cellular growth of HeLa cells”:

“As we validated that CRISPR-iCas9 is a suitable system to perturb the tRNA expression level, we conducted a CRISPR- targeted tRNA variants competition experiment among our designed library of 20 sgRNAs, to evaluate how reduction in tRNA levels affects the proliferation of HeLa cells”.

– Subsection “The transition from proliferative to cell cycle arrest state requires a unique set of tRNA families”:

“Having established the differential roles of tRNAs for cellular proliferation, we turned to examine their essentiality in response to cell cycle arresting conditions.”

8) Several issues were raised apropos statistical analyses. In Figure 3C and D, to assess whether tested variables are truly independent, the authors should use a linear regression modelling Relative fitness ~ tRNA expression (in C) and Relative fitness ~ fraction CRISPR targeted tRNAs (in D).

As the reviewers suggested, we computed a linear regression for Figure 3C (Relative fitness ~ tRNA expression), and for other figures showing a linear relationship as well. The model statistics are mentioned in subsection “Proliferation tRNAs are more essential than differentiation tRNAs for cellular growth of HeLa cells”.

9) In addition, it is not clear why is z-transformation applied in Figure 5E? The heatmap summarizes tRNA essentiality, which in Figure 3 and Figure 5C, is depicted using an untransformed log2FC. Using z-transformed and untransformed values to estimate the same effects was thought not to be advisable.

We thank the reviewers for drawing our attention to this unnecessary transformation. We changed the values in Figure 5E to their untransformed version.

10) Finally, the authors should also include the number of biological replicates, types of statistical tests and their outcomes in each figure where applicable, as in some cases these are missing.

We added these technical parameters, such as biological replicates and statistics in the legends of the main and supplementary figures.

11) Several statements were found not to be adequately supported by the data. For example, the statement: "our results show that some tRNAs are essential specifically for cancerous cells and not in differentiated cells.… (and the next sentence)", was found not to be supported by the presented data. To this end, the authors are advised either to provide data corroborating these conclusions or to tone down their statements.

As the reviewers suggested, we rephrased statements as follows: “CRISPR-editing of specific tRNAs was found here to have a close-to-immediate halt of proliferation, even in the extreme case of HeLa cells. Crucially, our results show that some tRNAs are essential for fast growing cells, while slow growing cells, like most of the normal cells in the human body, are less depended on these tRNAs. Such selective essentiality is important as it may suggest that targeting these tRNAs in a mixture of healthy and cancerous cells may affect mainly cancer.” Discussion section.

12) Also, in the Discussion section, given that this work is the first in systematically knocking out tRNA gene families, some comment on the potential and limitations of the method appears to be warranted.

We added a paragraph discussing the challenges and potential of CRISPR-targeting of multi-gene families in the Discussion section.